# RANDOM FEATURE MODELS WITH LEARNABLE ACTIVATION FUNCTIONS

## ABSTRACT

Current random feature models typically rely on fixed activation functions, limiting their ability to capture diverse patterns in data. To address this, we introduce the Random Feature model with Learnable Activation Functions (RFLAF), a novel model that significantly enhances the expressivity and interpretability of traditional random feature (RF) models. We begin by studying the RF model with a single radial basis function, where we discover a new kernel and provide the first theoretical analysis on it. By integrating the basis functions with learnable weights, we show that RFLAF can represent a broad class of random feature models whose activation functions belong in $C_c(\mathbb{R})$. Theoretically, we prove that the model requires only about twice the parameter number compared to a traditional RF model to achieve the significant leap in expressivity. Experimentally, RFLAF demonstrates two key advantages: (1) it performs better across various tasks compared to traditional RF model with the same number of parameters, and (2) the optimized weights offer interpretability, as the learned activation function can be directly inferred from these weights. Our model paves the way for developing more expressive and interpretable frameworks within random feature models.

## 1 INTRODUCTION

Kernel methods are powerful tools for solving nonlinear learning problems by leveraging kernel functions to implicitly map data into high-dimensional spaces. However, they can be computationally intensive and lack scalability with large datasets. The random feature (RF) model, introduced by Rahimi & Recht (2008b), offers a solution by approximating kernel functions with a finite number of random features, allowing the application of linear algorithms while maintaining the kernel's essence (Li et al., 2021; Liu et al., 2021).

Despite their advantages, random feature models typically use a fixed activation function, limiting their adaptability during data fitting. This rigidity prevents the model from automatically searching for activation functions for optimal performance across various tasks. Meanwhile, recent work such as Kolmogorov-Arnold Networks (KANs) (Liu et al., 2024b) has witnessed the powerful expressivity and interpretability of the structure of learnable activation functions. Therefore, it is natural to consider enhancing random feature models by incorporating trainable activation functions.

Current methods involve parametrizing learnable activation functions with splines(Liu et al., 2024b; Fakhoury et al., 2022; Bohra et al., 2020; Aziznejad & Unser, 2019). However, the piecewise definition of splines precludes the extraction of a closed-form analytic kernel, hindering the derivation of theoretical insights. Moreover, models equipped with splines may face difficulty with efficient convergence. For instance, KANs uses LBFGS with line search instead of common gradient descent method to boost convergence.

In this paper, we propose the Random Feature models with Learnable Activation Functions (RFLAF), a innovative random feature model that parametrizes learnable activation functions using weighted sums of radial basis functions (RBFs) (Section 2). RBFs offer several benefits: the regularity of RBFs allows for the derivation of the kernel's analytic form, enabling further theoretical analysis. In addition, RBFs and their derivatives are easily computed, which may facilitate the convergence of the model. This integration aims to improve the expressivity of the class of random feature models, while ensuring simple implementation and optimization of the model. Moreover, the introduction of trainable activation functions naturally adds interpretability to the model. Our

study offers a comprehensive analysis of the model, covering both its theoretical foundations and the empirical validations. Our contributions are summarized as follows.

- We identify an unexplored kernel through studying RFLAF of the special case activated by a single RBF. We provide the first result on the analytic form of this kernel, and investigate its representation and approximation characteristics. We demonstrate that a single RBF as activation does not necessarily result in stronger expressivity of the RF model (Section 3).

- For general RFLAF, we develop rigorous analyses on approximation and generalization. Our theories guarantee that our model boosts its representational capacity at a little cost of less than double parameter number (Section 4), and the number of random features only need to scale with the square root of the sample size (Section 5).

- Experimental results verify our theoretical findings (Section 6). Our model not only surpasses random feature models of other activation functions across different tasks, but is also simple to implement and shows rapid convergence rate with standard gradient descent optimizers. RFLAF successfully reveal the unknown activation functions through learning, presenting superior interpretability.

The rest of the paper is organized as follows. Section 2 outlines the basics of random feature models and formally introduces our model. Theoretical analyses for models with the single and combined RBF activations are provided in Sections 3 and 4 respectively. Section 5 discusses guarantees on sample complexity, and Section 6 presents experimental results to validate our theories.

## 1.1 RELATED WORK

**Random feature (RF) models.** Random feature model (Rahimi & Recht, 2008b; 2007) is initially motivated by the fact that randomization is computationally cheaper than optimization (Amit & Geman, 1997; Moosmann et al., 2006). By virtue of the relations between a kernel and its Fourier spectral density, random features act as a technique to scale up kernel methods (Lopez-Paz et al., 2014; Sun et al., 2018; Jacot et al., 2018; Arora et al., 2019b; Zandieh et al., 2021; Du et al., 2019; Zambon et al., 2020; Choromanski et al., 2020; Peng et al., 2021). When viewed as a class of two-layer neural networks with fixed weights in the first layer, random feature models provide deep insights for partly understanding deep neural networks (Cao & Gu, 2019; Arora et al., 2019a; Mei et al., 2022; Chizat & Bach, 2020). Research effort has been devoted to deriving approximation and generalization bounds with respect to random feature number and sample size (Sutherland & Schneider, 2015; Rudi & Rosasco, 2017). Li et al. (2021) contribute to a unified analysis of random Fourier features.

**Learnable activation functions.** Several work proposes the notion of learnable activation function and attempts to combine it into the structure of neural network. Activation functions are parametrized in a continous way as splines (Liu et al., 2024b;a; Fakhoury et al., 2022; Bohra et al., 2020; Aziznejad & Unser, 2019), polynomials (Goyal et al., 2019), sigmoid linear unit (Ramachandran et al., 2017) and neural network (Zhang et al., 2022), or in a dicrete way (Bingham & Miikkulainen, 2022). The similar notion of RBF network introduced by Lowe & Broomhead (1988) is fundamentally distinct from our model. In our work, RBFs are used for universal approximation, whereas the RBF network applies them for functional interpolation.

## 2 PRELIMINARIES

### 2.1 BASICS ON RANDOM FEATURE MODELS

In this section, we provide some foundations of random feature models (Rahimi & Recht, 2007; 2008a) related to our work.

Given a function $\sigma(x; w) : \mathcal{X} \times \mathcal{W} \to \mathbb{R}$. Let $\mu$ be a probability measure on $\mathcal{W}$. The class of infinite-width random feature model is defined as

$$\mathcal{F} = \left\{ f : f(x) = \int_{\mathcal{W}} \sigma(x; w) v(w) \mu(dw), v \in \mathcal{H}_{\mathcal{W}} \right\},$$

where $\mathcal{H}_{\mathcal{W}} = \left\{ v(w) : \int_{\mathcal{W}} v(w)^2 \mu(dw) < \infty \right\}$ is a Hilbert space with norm $\|v\|_{\mathcal{H}_{\mathcal{W}}}^2 = \int_{\mathcal{W}} v(w)^2 \mu(dw)$ and inner product $\langle v, u \rangle_{\mathcal{H}_{\mathcal{W}}} = \int_{\mathcal{W}} v(w)^\top u(w) \mu(dw)$. Furthermore, $\mathcal{F}$ is endowed with a norm $\| \cdot \|_{\mathcal{F}}$ and the inner product $\langle \cdot, \cdot \rangle_{\mathcal{F}}$:

$$\|f\|_{\mathcal{F}} = \inf_{f = \langle v, \sigma(\cdot) \rangle_{\mathcal{H}_{\mathcal{W}}}} \|v\|_{\mathcal{H}_{\mathcal{W}}}, \quad \langle f, g \rangle_{\mathcal{F}} = \frac{\|f + g\|_{\mathcal{F}}^2 - \|f - g\|_{\mathcal{F}}^2}{4}.$$

Besides, we define the corresponding reproducing kernel $K : \mathcal{X} \times \mathcal{X} \to \mathbb{R}$ as

$$K(x, y) = \int_{\mathcal{W}} \sigma(x; w) \sigma(y; w) \mu(dw).$$

Define the RKHS induced by this kernel as $\mathcal{H}_K$ with corresponding norm $\| \cdot \|_{\mathcal{H}_K}$ and the inner product $\langle \cdot, \cdot \rangle_{\mathcal{H}_K}$. Generally (Bai & Lee, 2019), for any feature map $\phi : \mathcal{X} \to \mathcal{H}$ (where $\mathcal{H}$ is a Hilbert space) that induces the kernel $K$, i.e., $K(x, y) = \langle \phi(x), \phi(y) \rangle_{\mathcal{H}}$, we have that for any function $f$,

$$\|f\|_{\mathcal{H}_K} = \inf_{f = \langle \mathbf{u}, \phi(\cdot) \rangle_{\mathcal{H}}} \|\mathbf{u}\|_{\mathcal{H}},$$

which indicates the equivalence among different feature maps that generate the same kernel.

Finally, we have the following proposition according to (Minh et al., 2006).

**Proposition 2.1** *Given the above definition of $\mathcal{F}$ and $\mathcal{H}_K$, we have that $(\mathcal{F}, \| \cdot \|_{\mathcal{F}}) = (\mathcal{H}_K, \| \cdot \|_{\mathcal{H}_K})$.*

### 2.2 Parametrization of activation functions and finite-width approximation

Standard random feature models consider the case where the activation function $\sigma$ is a fixed univariate function such as ReLU, and $\sigma(x; w) = \sigma(w^\top x)$. In this work, we broaden the target function class where $\sigma$ can be any function in $C_c(\mathbb{R})$, namely the continuous functions with compact support.

Let $x \in \mathbb{R}^d$, and $w \sim \mathcal{N}(0, I_d)$. For technical convenience, we assume $\sigma : \mathbb{R} \to \mathbb{R}$ and $v : \mathbb{R}^d \to \mathbb{R}$ to be Lipschitz continous. Suppose that $\sigma$ is supported on a bounded closed interval $\mathcal{K} \subseteq \mathbb{R}$. We formally define the target function class as

$$\mathcal{F}_{\mathcal{K}, L, L_v} := \left\{ f : f(x) = \mathbb{E}_{w \sim \mathcal{N}(0, I_d)} \left[ \sigma(w^\top x) v(w) \right], \sigma \in C_c(\mathcal{K}), \|\sigma\|_{\mathrm{Lip}} \leq L, \|v\|_{\mathrm{Lip}} \leq L_v \right\},$$
(2.1)

where $\| \cdot \|_{\mathrm{Lip}}$ denotes the Lipschitz constant of a function.

Suppose the target function $f = \mathbb{E} \left[ \sigma(w^\top x) v(w) \right] \in \mathcal{F}_{\mathcal{K}, L, L_v}$. The motivations of our model are twofold. In the first step, we consider using radial basis functions as basis for approximating arbitrary activation fucntions. Assume a list of radial basis functions

$$\left\{ B_i(x) = \exp \left( -\frac{(x - c_i)^2}{2h_i^2} \right) \right\}_{i \in [N]}$$

with centers $c_i$ and widths $h_i$ set in prior. By integrating $B_i$'s with learnable weights, we propose

$$\tilde{\sigma}(x) := \sum_{i=1}^{N} a_i B_i(x)$$

as the learnable activation function, where $a_i$ are learnable parameters. We expect to have that $\tilde{f}(x) := \mathbb{E}_{w \sim \mathcal{N}(0, I_d)} \left[ \tilde{\sigma}(w^\top x) v(w) \right] \approx f(x)$. In the second step, we approximate $\tilde{f}(x)$ with the finite-width random feature model $\frac{1}{M} \sum_{m=1}^{M} \tilde{\sigma}(w_m^\top x) v(w_m) \approx \mathbb{E}_{w \sim \mathcal{N}(0, I_d)} \left[ \tilde{\sigma}(w^\top x) v(w) \right]$. Assume $\{w_m\}_{m=1}^{M} \overset{i.i.d}{\sim} \mathcal{N}(0, I_d)$ are sampled. The random feature model with learnable activation functions is defined as

$$\hat{f}(x; \boldsymbol{a}, \boldsymbol{v}) := \frac{1}{M} \sum_{m=1}^{M} \sum_{i=1}^{N} a_i B_i(w_m^\top x) v_m,$$
(2.2)

where $\boldsymbol{a} = (a_1, ..., a_N) \in \mathbb{R}^N, \boldsymbol{v} = (v_1, ..., v_M) \in \mathbb{R}^M$ are learnable parameters. In the following sections, we will focus on the approximation error between $\hat{f}$ and $f$ and the sample complexity of learning.

## 3 RANDOM FEATURE MODELS WITH A SINGLE RADIAL BASIS FUNCTION

We first study the random feature model with a single radial basis function, which is a special case of our model with $N = 1$. We establish the mathematical foundations of the kernel induced by the model, including its analytic expression, its representational abllity and the approximation bounds, and conclude this section by discussing its limitations in expressivity.

The target function of interest admits representation

$$\varphi(x) = \mathbb{E}_{w \sim \mathcal{N}(0, I_d)} \left[ B(w^\top x) v(w) \right], \tag{3.1}$$

where the activation function $B(x) = \exp\left(-(x-c)^2/(2h^2)\right)$ is a radial basis function with center $c$ and width $h$. The corresponding reproducing kernel is

$$K(x, x') := \mathbb{E}_{w \sim \mathcal{N}(0, I_d)} \left[ B(w^\top x) B(w^\top x') \right]. \tag{3.2}$$

The first result presents the explicit expression of the kernel.

**Theorem 3.1** *For any $x, x' \in \mathbb{R}^d$, we have that*

$$K(x, x') = \frac{h^2}{\sqrt{(h^2 + \|x\|^2)(h^2 + \|x'\|^2) - \langle x, x' \rangle^2}} \exp\left( - \frac{c^2}{2} \cdot \frac{(h^2 + \|x\|^2) + (h^2 + \|x'\|^2) - 2\langle x, x' \rangle}{(h^2 + \|x\|^2)(h^2 + \|x'\|^2) - \langle x, x' \rangle^2} \right). \tag{3.3}$$

The proof is contained in Appendix B.1. Note that the kernel is similar to but different from the specific category called dot-product kernels (Smola et al., 2000). Hence, theoretical analysis of this kernel is lacking in the literature.

Without loss of generality, we assume that $\|x\|_2 = \|x'\|_2 = 1$, so that $r = \langle x, x' \rangle \in [-1, 1]$. Then the kernel degenerates into a rotation-invariant kernel (Liu et al., 2021). We slightly abuse the notation and define the univariate function $K(r)$ to be the rotation-invariant form of the kernel (3.3).

$$K(r) := \frac{h^2}{\sqrt{(1 + h^2)^2 - r^2}} \exp\left( - \frac{c^2}{1 + h^2 + r} \right). \tag{3.4}$$

To the best of our knowledge, this work is the first study on kernel (3.4). Hence, we develop rigorous mathematical results of this kernel from scratch. To obtain the representation theorem of the kernel, we consider its Taylor expansion.

**Theorem 3.2** *The rotation-invariant kernel $K(r)$ has Taylor expansion as*

$$K(r) = e^{-p} \frac{h^2}{1 + h^2} \sum_{n=0}^{\infty} \frac{R_n(p)}{n!(1 + h^2)^n} r^n, \tag{3.5}$$

*where $p = \frac{c^2}{1+h^2} \in [0, +\infty)$, and the polynomials are $R_n(x) = \begin{cases} P_k^2(x), & n = 2k, \\ x Q_k^2(x), & n = 2k + 1, \end{cases}$*

$$P_k(x) = \sum_{i=0}^{k} (-1)^{k-i} \frac{(2k-1)!!}{(2i-1)!!} \cdot \binom{k}{i} x^i, \quad Q_k(x) = \sum_{i=0}^{k} (-1)^{k-i} \frac{(2k+1)!!}{(2i+1)!!} \cdot \binom{k}{i} x^i.$$

*Therefore, the feature mapping with respect to the kernel (3.4) is*

$$\phi(x) = \left( \frac{h e^{-\frac{p}{2}} R_n^{\frac{1}{2}}(p)}{\sqrt{n!(1 + h^2)^{n+1}}} x^{\otimes n} \right)_{n=0}^{\infty}.$$

The proof is contained in Appendix B.2. Define the represented function class as

$$\mathcal{H}_{c,h} = \left\{ \varphi : \mathbb{S}^{d-1} \to \mathbb{R} \,\middle|\, \varphi(x) = \sum_{n=0}^{\infty} \langle F_n, x^{\otimes n} \rangle, \quad D_{c,h}(\varphi) < \infty \right\},$$

where $F_n \in \mathbb{R}^{d^n}$ and

$$D_{c,h}(\varphi) := \frac{e^p}{h^2} \sum_{n=0}^{\infty} \frac{n!(1+h^2)^{n+1}}{R_n(c^2/(1+h^2))} \|F_n\|_{\mathrm{Fr}}^2.$$

Then we have the following representation theorem.

**Corollary 3.3** *For any $f \in \mathcal{H}_{c,h}$, there exists $v : \mathbb{R}^d \to \mathbb{R}$ such that $f(x) = \mathbb{E}_w \left[ B(w^\top x) v(w) \right]$ and $\mathbb{E}_w \left[ v(w)^2 \right] \leq D_{c,h}(f)$, where $w \sim \mathcal{N}(0, I_d)$ and $B(x) = \exp\left( -\frac{(x-c)^2}{2h^2} \right)$.*

The proof is contained in Appendix B.3. Next, we aim to approximate (3.1) using the finite-width random feature model

$$\hat{\varphi}(x) = \frac{1}{M} \sum_{m=1}^{M} B(w_m^\top x) v_m.$$

Recall that in (2.1), we assume a mild condition that $v$ is $L_v$-Lipschitz continuous. Because $|v(w)|^2 \leq (|v(\mathbf{0})| + L_v \|w\|)^2 \leq 2v(\mathbf{0})^2 + 2L_v^2 \|w\|^2$. By setting $R = \sqrt{2L_v^2 d + 2|v(\mathbf{0})|^2}$, we have $\mathbb{E}_{w \sim \mathcal{N}(0, I_d)} \left[ v(w)^2 \right] \leq R^2$. Now, we present the approximation error below.

**Theorem 3.4** *Let $v$ be $L_v$-Lipschitz and $R = \sqrt{2L_v^2 d + 2|v(\mathbf{0})|^2}$. Suppose that $\{w_m\}_{m=1}^{M} \overset{i.i.d}{\sim} \mathcal{N}(0, I_d)$, then with probability of at least $1 - \delta$, there exists $\{v_m\}_{m=1}^{M}$ such that*

$$\mathbb{E}_x |\hat{\varphi}(x) - \varphi(x)| \leq \frac{18R\sqrt{\log(4/\delta)}}{\sqrt{M}},$$

*and*

$$\frac{1}{M} \sum_{m=1}^{M} v_m^2 \leq 49R^2 \log(2/\delta),$$

*where we assume $\delta < 1/2$. Note that the inequalities hold for whatever distribution $x$ are sampled from.*

The proof is contained in Appendix B.4. The proof of Theorem 3.4 is not trivial, because the concentration property of $|\hat{\varphi}(x) - \varphi(x)|$ may not be uniform over $x$. We use some techiniques to circumvent this problem.

Implied by Corollary 3.3, the represented function $f$ must have coefficients $\|F_n\|^2 = o(1/n^n)$, indicating a narrow function class. Hence, using a single RBF as the activation function does not necessarily lead to a leap in expressivity of the RF model. The key step is to combine the RBFs with learnable weights. The mechanism of learnable activation functions results in evidently enhanced expressivity of RF models, as we will demonstrate in the next section.

## 4 RANDOM FEATURE MODELS WITH LEARNABLE ACTIVATION FUNCTIONS

At the beginning of this section, we recall the Gaussian universal approximation theorem in the sense of $C_\infty$ norm in (Bacharoglou, 2010; Nestoridis & Stefanopoulos, 2007). In the notations hereafter, we denote $c_i$ and $h_i$ as the parameters of the function $B_i(x) = \exp\left(-(x-c_i)^2/(2h_i^2)\right)$.

**Gaussian Universal Approximation Theorem (Gaussian UAT).** Suppose the target function $\sigma(x)$ is a continuous function with compact support $\mathcal{K}$. For any $\epsilon > 0$, there exists $N > 0$ and $\{h_i, c_i, a_i\}_{i=1}^{N}$ such that

$$\left\| \sigma(x) - \sum_{i=1}^{N} a_i B_i(x) \right\|_\infty < \epsilon. \tag{4.1}$$

Inspired by the theorem, for any target function $f^* \in \mathcal{F}_{\mathcal{K}, L, L_v}$, we aim to first approximate it with $\tilde{f}(x) := \mathbb{E}_w \left[ \sum_{i=1}^{N} a_i B_i(w^\top x) v(w) \right]$, in which $\{a_i\}_{i \in [N]}$ are learnable and $\{c_i, h_i\}_{i \in [N]}$ are set in prior. To describe $c_i$ and $h_i$ precisely, we partition the support set $\mathcal{K}$ of $\sigma$.

Let the grid number be $N$. We define the grid points as $y_0 = \inf_{x \in \mathcal{K}} x$, $y_N = \sup_{x \in \mathcal{K}} x$ and $y_i = y_0 + \frac{i}{N}(y_N - y_0)$ for $1 \le i \le N - 1$. The grid size then is $|\mathcal{K}|/N$. The diameter of the support is $|\mathcal{K}| := y_N - y_0$. Because $\sigma$ is continuous over the compact set $\mathcal{K}$, it is also bounded. Hence, $\|\sigma\|_\infty < \infty$. Our first result measures the approximation error between $f^*$ and $\tilde{f}$ with respect to the choice of $h_i$ and grid size.

**Proposition 4.1** *Suppose $f^* \in \mathcal{F}_{\mathcal{K},L,L_v}$ with activation function $\sigma$. For any $\epsilon > 0$, by setting*

$$h \le \frac{\epsilon}{4\sqrt{2}LR\sqrt{\log \frac{16\|\sigma\|_\infty R}{\epsilon}}}, \quad \frac{|\mathcal{K}|}{N} \le \frac{\epsilon h \sqrt{\pi e}}{16\sqrt{2}\|\sigma\|_\infty R \log\left(\frac{8\|\sigma\|_\infty |\mathcal{K}| R}{\sqrt{2\pi \epsilon h^2}}\right)} \wedge \frac{\epsilon}{4LR},$$

*and $h_i = h$, $c_i \in [y_{i-1}, y_i]$, there exists $\{a_i\}_{i=1}^N$ such that*

$$\left\| \tilde{f}(x) - f^*(x) \right\|_\infty \le \epsilon, \quad \left\| \sigma(x) - \sum_{i=1}^N a_i B_i(x) \right\|_\infty < \epsilon/R, \quad (4.2)$$

*and*

$$\sum_{i=1}^N |a_i| \le \frac{\|\sigma\|_\infty |\mathcal{K}|}{\sqrt{2\pi}h}, \quad \sum_{i=1}^N |a_i|^2 \le \frac{\|\sigma\|_\infty^2 |\mathcal{K}|^2}{2\pi h^2 N}.$$

The proof of Proposition 4.1 is contained in Appendix C.1, the techniques of which are similar to those in (Bacharoglou, 2010; Nestoridis & Stefanopoulos, 2007). Note that the $L_2$ bound implies the $L_1$ bound because $\sum_{i=1}^N |a_i| \le \sqrt{\sum_{i=1}^N |a_i|^2}\sqrt{N}$. Hence, we can omit the $L_1$ bound hereafter.

Next, we attempt to approximate $\tilde{f}$ with finite-width random feature model. Denote

$$\varphi_i(x) = \mathbb{E}_{w \sim \mathcal{N}(0, I_d)}\left[B_i(w^\top x)v(w)\right], \quad \hat{\varphi}_i(x) = \frac{1}{M}\sum_{m=1}^M B_i(w_m^\top x)v_m.$$

Then

$$\tilde{f}(x) = \sum_{i=1}^N a_i \varphi_i(x), \quad \hat{f}(x) = \sum_{i=1}^N a_i \hat{\varphi}_i(x).$$

Our second result measures the approximation error between $f^*$ and $\hat{f}$ in the sense of $L_1$ norm.

**Theorem 4.2** *Suppose $f^* \in \mathcal{F}_{\mathcal{K},L,L_v}$ with activation function $\sigma$. For all $\epsilon > 0$, under the parameter settings of Proposition 4.1, let $\{w_m\}_{m=1}^M \overset{i.i.d}{\sim} \mathcal{N}(0,1)$, then with probability of at least $1 - \delta$, there exists $\{a_i\}_{i=1}^N$ and $\{v_m\}_{m=1}^M$ such that*

$$\mathbb{E}_x \left| \hat{f}(x) - f^*(x) \right| \le \frac{18(\|\sigma\|_\infty R + \epsilon)\sqrt{\log(4/\delta)}}{\sqrt{M}} + \epsilon,$$

*and*

$$\sum_{i=1}^N a_i^2 \le \frac{\|\sigma\|_\infty^2 |\mathcal{K}|^2}{2\pi h^2 N}, \quad \sum_{m=1}^M v_m^2 \le 49MR^2 \log\left(\frac{2}{\delta}\right),$$

*where we assume $\delta < 1/2$.*

The proof of Theorem 4.2 is contained in Appendix C.2. Theorem 4.2 indicates that to obtain $O(\epsilon)$ approximation error, the model requires $M = \Theta(1/\epsilon^2)$. In the meantime, Proposition 4.1 indicates that $1/h = \widetilde{\Theta}(1/\epsilon)^1$ and $N = \widetilde{\Theta}(1/\epsilon h)$. Hence, $N = \widetilde{\Theta}\left(1/\epsilon^2\right) = \widetilde{\Theta}(M)$. The number of grid points $N$ should scale with approximately the same order of $M$. When implementing the models, one can typically set $N \lesssim M$ (see section 6). To compare, the RFLAF model has $M + N$ parameters, $N$ more than a standard RF model of the same width. Setting $N \lesssim M$ as suggested before, our model gains enhanced expressivity with just a twofold increase in parameter numbers.

---

[1]$\widetilde{\Theta}(\cdot)$ stands for $\Theta(\cdot)$ but hides the logarithmic terms

## 5    GENERALIZATION BOUNDS AND SAMPLE COMPLEXITY OF LEARNING

To complete the theoretical analysis of the model, we introduce the generalization bounds of learning in this section.

Suppose the data distribution is $\mathsf{P}$ and the samples are $S = \{(x_i, y_i)\}_{i=1}^n \overset{i.i.d.}{\sim} \mathsf{P}$. Suppose the loss function $\ell(\hat{y}, y)$ is $\rho$-Lipschitz in $\hat{y}$ and $|\ell(0, y)| \leq \rho$ for any $y$. The population risk and the empirical risk are defined respectively as

$$L_D(f) := \mathbb{E}_{x,y \sim \mathsf{P}}[\ell(f(x), y)], \quad L_S(f) := \frac{1}{n} \sum_{i=1}^n \ell(f(x_i), y_i).$$

The minimizer of the population risk is

$$f^* := \underset{f \in \mathcal{F}_{\mathcal{K},L,L_v}}{\operatorname{argmin}} L_D(f). \tag{5.1}$$

Under the setting of Theorem 4.2, suppose $\{w_m\}_{m=1}^M \overset{i.i.d}{\sim} \mathcal{N}(0, I_d)$ are sampled, $h$, $N$ and $\{c_i\}_{i \in [N]}$ are fixed, we aim at learning the parameters $V = (\boldsymbol{a}, \boldsymbol{v}) = (a_1, ..., a_N, v_1, ..., v_M)$ in $f_V(x) := \frac{1}{M} \sum_{m=1}^M \sum_{i=1}^N a_i B_i(w_m^\top x) v_m$. Guided by Theorem 4.2, the constrained set is set to be

$$\mathcal{V} := \left\{ V = (\boldsymbol{a}, \boldsymbol{v}) \in \mathbb{R}^N \times \mathbb{R}^M : \|\boldsymbol{a}\|_2 \leq \frac{\|\sigma\|_\infty |\mathcal{K}|}{h\sqrt{2\pi N}}, \quad \|\boldsymbol{v}\|_2 \leq 7R\sqrt{M \log\left(\frac{2}{\delta}\right)} \right\}. \tag{5.2}$$

Denote $f_{\mathcal{V}} = \{f_V\}_{V \in \mathcal{V}}$. The minimizer of the empirial risk is

$$f_S := \underset{f_V \in f_{\mathcal{V}}}{\operatorname{argmin}} L_S(f_V). \tag{5.3}$$

**Theorem 5.1** *Under the conditions and parameter settings of $h, N, \{c_i\}_{i=1}^N$ in Theorem (4.2), let $f^*$ and $f_S$ be the minimizers of the population risk and the empirical risk in Eq. (5.1) and (5.3) respectively. For all $\epsilon > 0$, with probability of at least $1 - \delta$ over $\{w_m\}_{m=1}^M \overset{i.i.d}{\sim} \mathcal{N}(0, I_d)$ and $\{(x_i, y_i)\}_{i=1}^n \overset{i.i.d.}{\sim} \mathsf{P}$, the excess risk is bounded by*

$$L_D(f_S) - L_D(f^*) \leq \frac{\rho C \log(16/\delta)}{h\sqrt{n}} + \frac{\rho C \sqrt{\log(8/\delta)}}{\sqrt{M}} + \rho\epsilon,$$

*where $C = \max\{14\left(1 + 7\|\sigma\|_\infty |\mathcal{K}|R\right), 18(\|\sigma\|_\infty R + \epsilon)\}$, and we assume $\delta \leq 1/2$, $h \leq 1$.*

The proof is contained in Appendix D, which mainly boils down to estimating the Rademacher complexity of the function class induced by the constrained set (5.2).

Theorem 5.1 implies that to achieve $O(\epsilon)$ excess risk, it suffices to have the sample size $n$, the random feature number $M$ and the grid number $N$ to scale as

$$n = \widetilde{\Theta}\left(1/\epsilon^2 h^2\right), \quad M = \Theta\left(1/\epsilon^2\right), \quad N = \widetilde{\Theta}\left(1/\epsilon h\right).$$

Indicated by Proposition 4.1, we set $h$ such that $1/h = \widetilde{\Theta}(1/\epsilon)$. Hence, only $M = \widetilde{\Theta}(\sqrt{n})$ number of random features are required, matching the sharpest results on the number of features presented in (Li et al., 2021; Rudi & Rosasco, 2017) for random feature models of fixed activation. We will substantiate these findings through experimental results in the subsequent section.

## 6    NUMERICAL EXPERIMENTS

We choose target functions to be of the form $f(x) = \mathbb{E}_{w \sim \mathcal{N}(0, I_d)}\left[\sigma(w^\top x)v(w)\right]$ where $x \in \mathbb{R}^d$ and $d = 2$. We set $f_1, f_2, f_3$ with the corresponding $\sigma_1, \sigma_2, \sigma_3$ as

$$\sigma_1(x) = \sin(\pi x)\mathbf{1}_{[-1,1]}, \qquad \sigma_2(x) = \sin(\pi x)\mathbf{1}_{[0,1]},$$
$$\sigma_3(x) = -\sin(\pi(x + 0.5))\mathbf{1}_{[-1.5, -0.5]} + \sin(\pi(x - 0.5))\mathbf{1}_{[0.5, 1.5]},$$

and $v_i(w) = c_i \max\{b_1^\top w, b_2^\top w\}, i \in [3]$, where $b_1, b_2$ are two fixed vectors, and $c_i$ are set as to ensure that $\mathbb{E}_x |f_i(x)| \approx 1$.

For the RFLAF model, we set the random feature number $M = 1000$, the grid number $N = 400$, the support area of the learnable activation function $\mathcal{K} = [-2, 2]$, the centers of RBFs as the grid points of $\mathcal{K}$, and the width of RBFs as $h = 0.02$.

For comparison, we consider random feature models with a list of activation functions, the width of which are set as $M + N$ to ensure all the models have the same parameter number. We consider several common activation functions such as ReLU, Tanh. Specifically, we also consider a single RBF as the activation function, where RBF1 is $\exp(-x^2/(2 \cdot (0.5)^2))$ and RBF2 is $\exp(-(x - 1.5)^2/(2 \cdot (0.5)^2))$. We apply MSE loss with certain regularizers in the learning process. More details about the optimization setup can be found in appendix E.

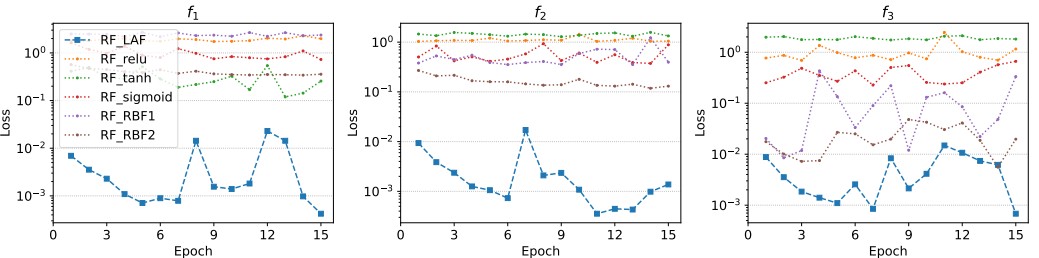

Figure 1: Test losses

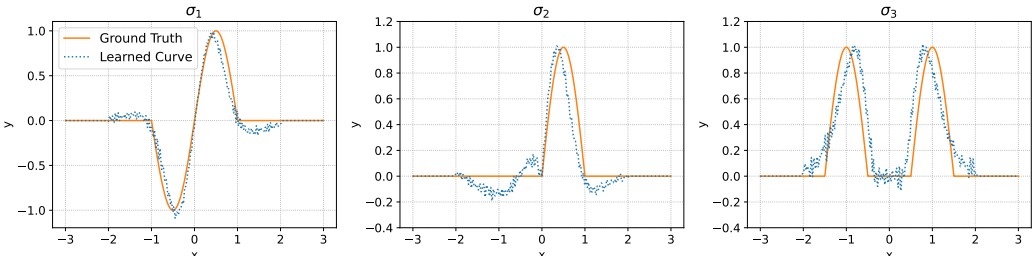

Figure 2: Learned activation functions

Experimental results show several advantages of our model:

**Enhanced expressivity.** Figure 1 shows that RFLAF significantly outperforms other models across the three tasks. RFLAF achieves consistently low level of losses, while other models fail to correctly learn the target functions. Specifically, RF models with RBF as a single activation do not learn all the target functions, demonstrating that the stronger expressivity of RFLAF comes from the universal approximation characteristics of the array of RBFs. We also find similar phenomena across a wider range of activation functions. The result is contained in appendix E.

**Emerged interpretability.** Figure 2 shows that RFLAF provides a level of interpretability concerning the structure of the activation function. By combining the RBFs with the optimized weights, we find that RFLAF successfully recover the activation functions of various forms, providing deeper insights into the data patterns. Trainable activation functions act as the core component in a recently proposed deep learning framework, Kolmogorov-Arnold Network (Liu et al., 2024b;a). Our findings highlight the potentials of trainable activation functions to provide interpretability from the perspective of random features.

**Easy implementation and fast convergence.** The implementation of our model is simple. It merely involves appending a list of modules consists of RBFs in PyTorch compared to an ordinary

RF model. In addition, a standard gradient descent optimizer such as Adam should suffice for fast convergence of our model. Although a wide range of functions serve as basis functions in UAT (Nestoridis & Stefanopoulos, 2007), we use RBFs instead of B-splines in (Liu et al., 2024b). The reasons are twofold: (1) RBFs are regular, simplifying the theoretical analysis, (2) RBFs and their derivatives are easily computed, facilitating the convergence process. In contrast, models with B-splines typically require LBFGS with line search for better convergence performance. Figure 1 provides evidence for the rapid convergence of our model, occurring within a duration of 5 epochs.

## 7 CONCLUSION

In this work, we propose the random feature model with learnable activation functions. We provide theoretical results on the approximation and generalization properties, and validate its superior expressivity and interpretability empirically. These findings deepen our comprehension of the benefits of learnable activation functions, and lay the groundwork for the advancement of more expressive and interpretable machine learning paradigms that incorporate such mechanism. Our work initiates an array of open problems for future work, including the derivation of tighter generalization bounds, the exploration of basis functions for more effective approximation and faster convergence, and the extension of learnable activation functions to a broader spectrum of statistical learning models.

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

## A    TECHNICAL TOOLS

### A.1    BASICS ON SUB-GAUSSIAN AND SUB-EXPONENTIAL RANDOM VARIABLES

**Definition A.1** *A random variable $Y$ is a sub-gaussian random variable if there exists $K > 0$ such that $\mathbb{E}\exp\left(Y^2/K^2\right) \leq 2$. Define the sub-gaussian norm as $\|Y\|_{\psi_2} := \inf\{K > 0 : \mathbb{E}\exp\left(Y^2/K^2\right) \leq 2\}$.*

**Definition A.2** *A random variable $Y$ is a sub-exponential random variable if there exists $K > 0$ such that $\mathbb{E}\exp\left(|Y|/K\right) \leq 2$. Define the sub-exponential norm as $\|Y\|_{\psi_1} := \inf\{K > 0 : \mathbb{E}\exp\left(|Y|/K\right) \leq 2\}$.*

**Lemma A.1** *If $Y$ is a sub-gaussian random variable, then $\|Y^2\|_{\psi_1} = \|Y\|_{\psi_2}^2$.*

The following properties of sub-gaussian random variable are stated in Proposition 2.5.2 in (Vershynin, 2018). For this paper to be self-contained, we also state them here with explicit constants.

**Lemma A.2** *Suppose $Y$ is a random variable.*

1. *If $Y$ is a sub-gaussian random variable, then $P(|Y| \geq \epsilon) \leq 2\exp\left(-\epsilon^2/\|Y\|_{\psi_2}^2\right)$.*

2. *If $P(|Y| \geq \epsilon) \leq 2\exp\left(-\epsilon^2/K^2\right)$, then $\|Y\|_{\psi_2} \leq \sqrt{2}K$.*

**Lemma A.3** *Suppose $Y$ is a random variable.*

1. *If there exist $K_0 > 0$ such that $\mathbb{E}e^{\lambda^2 Y^2} \leq e^{K_0^2\lambda^2}$ for all $\lambda^2 \leq 1/K_0^2$, then $Y$ is a sub-gaussian random variable with sub-gaussian norm $\|Y\|_{\psi_2} \leq K_0/\sqrt{\log 2} \leq \sqrt{2}K_0$.*

2. *If $Y$ is a sub-gaussian random variable, then $K_0 = 2\|Y\|_{\psi_2}$ such that $\mathbb{E}e^{\lambda^2 Y^2} \leq e^{K_0^2\lambda^2}$ for all $\lambda^2 \leq 1/K_0^2$.*

**Lemma A.4** *Suppose $Y$ is a random variable and $\mathbb{E}Y = 0$.*

1. *If $\mathbb{E}e^{\lambda^2 Y^2} \leq e^{K_0^2\lambda^2}$ for all $\lambda^2 \leq 1/K_0^2$, then $\mathbb{E}e^{\lambda Y} \leq e^{K_0^2\lambda^2}$ for all $\lambda \in \mathbb{R}$.*

2. *If $\mathbb{E}e^{\lambda Y} \leq e^{K_0^2\lambda^2}$ for all $\lambda \in \mathbb{R}$, then $\mathbb{E}e^{\lambda^2 Y^2} \leq e^{16K_0^2\lambda^2}$ for all $\lambda^2 \leq 1/16K_0^2$.*

For sums of independent sub-gaussian random variables, the Proposition 2.6.1 in (Vershynin, 2018) states that

**Lemma A.5** *Let $X_1, ..., X_M$ be independent copies of a sub-gaussian random variable $X$ and $\mathbb{E}X = 0$. Then*

$$\left\|\sum_{m=1}^{M} X_m\right\|_{\psi_2} \leq 4\sqrt{M}\,\|X\|_{\psi_2}.$$

We also state a concentration inequality for sums of independent sub-exponential random variables.

**Lemma A.6 (Bernstein's inequality (e.g., Theorem 2.8.1 in (Vershynin, 2018)))** *Let $X_1, ..., X_M$ be independent copies of a sub-exponential random variable $X$ and $\mathbb{E}X = 0$. Then, for every $t > 0$, we have*

$$P\left(\frac{1}{M}\sum_{m=1}^{M} X_m > t\right) \leq \exp\left(-\min\left\{\frac{Mt^2}{16\|X\|_{\psi_1}^2}, \frac{Mt}{4\|X\|_{\psi_1}}\right\}\right).$$

## B  DEFERRED PROOF IN SECTION 3

### B.1  PROOF OF THEOREM 3.1

**Proof.** For the simplicity of calculation, we denote

$$s := \|x\|_2, \quad s' := \|x'\|_2, \quad \rho = \frac{\langle x, x' \rangle}{\|x\|_2 \|x'\|_2}.$$

The statistical properties of Gaussian distribution indicate that

$$\mathbb{E}_{w \sim \mathcal{N}(0, I_d)} \left[ B(w^\top x) B(w^\top x') \right] = \mathbb{E}_{x \sim \mathcal{N}(0,1), z \sim \mathcal{N}(0, 1-\rho^2)} \left[ B(sx) B(s'(\rho x + z)) \right],$$

where $x, z$ are two independent Gaussian random variables. Then we do the calculation based on the latter expression. The calculation is quite complicated, so we illustrate it here in a detailed way.

$$\mathbb{E}_{x \sim \mathcal{N}(0,1), z \sim \mathcal{N}(0, 1-\rho^2)} \left[ B(sx) B(s'(\rho x + z)) \right]$$

$$= \frac{1}{\sqrt{2\pi}} \frac{1}{\sqrt{2\pi(1-\rho^2)}} \iint_{\mathbb{R}^2} \exp\left( -\frac{(sx-c)^2}{2h^2} \right) \exp\left( -\frac{(s'(\rho x + z) - c)^2}{2h^2} \right)$$

$$\exp\left( -\frac{x^2}{2} \right) \exp\left( -\frac{z^2}{2(1-\rho^2)} \right) dx dz$$

$$= \frac{1}{\sqrt{2\pi}} \frac{1}{\sqrt{2\pi(1-\rho^2)}} \iint_{\mathbb{R}^2} \exp\left( -\frac{1}{2h^2(1-\rho^2)} \left[ (1-\rho^2)(sx-c)^2 + (1-\rho^2)(s'\rho x - c + s'z)^2 \right. \right.$$

$$\left. \left. + (1-\rho^2)h^2 x^2 + h^2 z^2 \right] \right) dx dz$$

$$= \frac{1}{\sqrt{2\pi}} \frac{1}{\sqrt{2\pi(1-\rho^2)}} \iint_{\mathbb{R}^2} \exp\left( -\frac{1}{2h^2(1-\rho^2)} \left[ (1-\rho^2)(sx-c)^2 + (1-\rho^2)(s'\rho x - c)^2 + (1-\rho^2)(s'z)^2 \right. \right.$$

$$\left. \left. + 2(1-\rho^2)(s'\rho x - c)s'z + (1-\rho^2)h^2 x^2 + h^2 z^2 \right] \right) dx dz$$

$$= \frac{1}{\sqrt{2\pi}} \frac{1}{\sqrt{2\pi(1-\rho^2)}} \iint_{\mathbb{R}^2} \exp\left( -\frac{1}{2h^2(1-\rho^2)} \left[ (1-\rho^2)(sx-c)^2 + (1-\rho^2)(s'\rho x - c)^2 + (1-\rho^2)h^2 x^2 \right. \right.$$

$$\left. \left. + [(1-\rho^2)(s')^2 + h^2]z^2 + 2(1-\rho^2)(s'\rho x - c)s'z \right] \right) dx dz$$

$$= \frac{1}{\sqrt{2\pi}} \frac{1}{\sqrt{2\pi(1-\rho^2)}} \iint_{\mathbb{R}^2} \exp\left( -\frac{1}{2h^2(1-\rho^2)} \left[ (1-\rho^2)(sx-c)^2 + (1-\rho^2)(s'\rho x - c)^2 + (1-\rho^2)h^2 x^2 \right. \right.$$

$$\left. \left. + [(1-\rho^2)(s')^2 + h^2] \left[ z + \frac{(1-\rho^2)(s'\rho x - c)s'}{(1-\rho^2)(s')^2 + h^2} \right]^2 - \frac{(1-\rho^2)^2(s'\rho x - c)^2(s')^2}{(1-\rho^2)(s')^2 + h^2} \right] \right) dx dz$$

$$= \frac{1}{\sqrt{2\pi}} \frac{1}{\sqrt{2\pi(1-\rho^2)}}$$

$$\int_{\mathbb{R}} \exp\left( -\frac{(1-\rho^2)(sx-c)^2 + (1-\rho^2)(s'\rho x - c)^2 + (1-\rho^2)h^2 x^2 - \frac{(1-\rho^2)^2(s'\rho x - c)^2(s')^2}{(1-\rho^2)(s')^2 + h^2}}{2h^2(1-\rho^2)} \right) dx$$

$$\int_{\mathbb{R}} \exp\left( -\frac{\left[ z + \frac{(1-\rho^2)(s'\rho x - c)s'}{(1-\rho^2)(s')^2 + h^2} \right]^2}{2 \frac{h^2(1-\rho^2)}{(1-\rho^2)(s')^2 + h^2}} \right) dz$$

$$= \frac{1}{\sqrt{2\pi}} \frac{\sqrt{\frac{h^2(1-\rho^2)}{(1-\rho^2)(s')^2 + h^2}}}{\sqrt{(1-\rho^2)}} \int_{\mathbb{R}} \exp\left( -\frac{(1-\rho^2)(sx-c)^2 + (1-\rho^2)(s'\rho x - c)^2 + (1-\rho^2)h^2 x^2 - \frac{(1-\rho^2)^2(s'\rho x - c)^2(s')^2}{(1-\rho^2)(s')^2 + h^2}}{2h^2(1-\rho^2)} \right) dx.$$

To continue, we have

$$\mathbb{E}_{x\sim\mathcal{N}(0,1),z\sim\mathcal{N}(0,1-\rho^2)}\left[B(sx)B(s'(\rho x+z))\right]$$

$$=\frac{1}{\sqrt{2\pi}}\frac{h}{\sqrt{(1-\rho^2)(s')^2+h^2}}$$

$$\int_{\mathbb{R}}\exp\left(-\frac{(1-\rho^2)(sx-c)^2+(1-\rho^2)(s'\rho x-c)^2+(1-\rho^2)h^2x^2-\frac{(1-\rho^2)^2(s'\rho x-c)^2(s')^2}{(1-\rho^2)(s')^2+h^2}}{2h^2(1-\rho^2)}\right)dx$$

$$=\frac{1}{\sqrt{2\pi}}\frac{h}{\sqrt{(1-\rho^2)(s')^2+h^2}}$$

$$\int_{\mathbb{R}}\exp\left(-\frac{(sx-c)^2+(s'\rho x-c)^2+h^2x^2-\frac{(1-\rho^2)(s')^2(s'\rho x-c)^2}{(1-\rho^2)(s')^2+h^2}}{2h^2}\right)dx$$

$$=\frac{1}{\sqrt{2\pi}}\frac{h}{\sqrt{(1-\rho^2)(s')^2+h^2}}$$

$$\int_{\mathbb{R}}\exp\left(-\frac{[(1-\rho^2)(s')^2+h^2][(sx-c)^2+h^2x^2]+h^2(s'\rho x-c)^2}{2h^2[(1-\rho^2)(s')^2+h^2]}\right)dx$$

$$=\frac{1}{\sqrt{2\pi}}\frac{h}{\sqrt{(1-\rho^2)(s')^2+h^2}}$$

$$\int_{\mathbb{R}}\exp\left(-\frac{[(1-\rho^2)(s')^2+h^2][(s^2+h^2)x^2-2scx+c^2]+h^2[(s')^2\rho^2x^2-2s'\rho cx+c^2]}{2h^2[(1-\rho^2)(s')^2+h^2]}\right)dx$$

$$=\frac{1}{\sqrt{2\pi}}\frac{h}{\sqrt{(1-\rho^2)(s')^2+h^2}}$$

$$\int_{\mathbb{R}}\exp\left(-\frac{[h^4+(1-\rho^2)s^2(s')^2+h^2(s^2+(s')^2)]x^2}{2h^2[(1-\rho^2)(s')^2+h^2]}-\right.$$

$$\left.\frac{-2[(1-\rho^2)(s')^2s+h^2s+h^2s'\rho]cx+[(1-\rho^2)(s')^2+2h^2]c^2}{2h^2[(1-\rho^2)(s')^2+h^2]}\right)dx$$

$$=\frac{1}{\sqrt{2\pi}}\frac{h}{\sqrt{(1-\rho^2)(s')^2+h^2}}$$

$$\int_{\mathbb{R}}\exp\left(-\frac{[h^4+(1-\rho^2)s^2(s')^2+h^2(s^2+(s')^2)]\left(x-\frac{[(1-\rho^2)(s')^2s+h^2s+h^2s'\rho]c}{h^4+(1-\rho^2)s^2(s')^2+h^2(s^2+(s')^2)}\right)^2}{2h^2[(1-\rho^2)(s')^2+h^2]}-\right.$$

$$\left.\frac{-\frac{[(1-\rho^2)(s')^2s+h^2s+h^2s'\rho]^2c^2}{h^4+(1-\rho^2)s^2(s')^2+h^2(s^2+(s')^2)}+[(1-\rho^2)(s')^2+2h^2]c^2}{2h^2[(1-\rho^2)(s')^2+h^2]}\right)dx$$

$$=\frac{h\sqrt{(1-\rho^2)(s')^2+h^2}}{\sqrt{h^4+(1-\rho^2)s^2(s')^2+h^2(s^2+(s')^2)}}\frac{h}{\sqrt{(1-\rho^2)(s')^2+h^2}}$$

$$\exp\left(\frac{\frac{[(1-\rho^2)(s')^2s+h^2s+h^2s'\rho]^2c^2}{h^4+(1-\rho^2)s^2(s')^2+h^2(s^2+(s')^2)}-[(1-\rho^2)(s')^2+2h^2]c^2}{2h^2[(1-\rho^2)(s')^2+h^2]}\right)$$

$$=\frac{h^2}{\sqrt{(h^2+s^2)(h^2+(s')^2)-\rho^2s^2(s')^2}}\exp\left(\frac{\frac{[(1-\rho^2)(s')^2s+h^2s+h^2s'\rho]^2c^2}{h^4+(1-\rho^2)s^2(s')^2+h^2(s^2+(s')^2)}-[(1-\rho^2)(s')^2+2h^2]c^2}{2h^2[(1-\rho^2)(s')^2+h^2]}\right).$$

For the exponential term, we calculate as follows.

$$\frac{\frac{[(1-\rho^2)(s')^2 s + h^2 s + h^2 s'\rho]^2 c^2}{h^4 + (1-\rho^2)s^2(s')^2 + h^2(s^2 + (s')^2)} - [(1-\rho^2)(s')^2 + h^2 + h^2]c^2}{2h^2[(1-\rho^2)(s')^2 + h^2]}$$

$$= -\frac{c^2}{2h^2}\left(1 - \frac{[(1-\rho^2)(s')^2 s + h^2 s + h^2 s'\rho]^2 - h^2[h^4 + (1-\rho^2)s^2(s')^2 + h^2(s^2 + (s')^2)]}{[(1-\rho^2)(s')^2 + h^2][h^4 + (1-\rho^2)s^2(s')^2 + h^2(s^2 + (s')^2)]}\right)$$

$$= -\frac{c^2}{2h^2}\left(1 - \frac{[((1-\rho^2)(s')^2 + h^2)s + h^2 s'\rho]^2 - h^2[((1-\rho^2)(s')^2 + h^2)s^2 + h^2(h^2 + (s')^2)]}{[(1-\rho^2)(s')^2 + h^2][(h^2 + s^2)(h^2 + (s')^2) - \rho^2 s^2(s')^2]}\right)$$

$$= -\frac{c^2}{2h^2}\left(1 - \frac{[((1-\rho^2)(s')^2 + h^2)^2 s^2 + h^4(s')^2\rho^2 + 2((1-\rho^2)(s')^2 + h^2)h^2 ss'\rho]}{[(1-\rho^2)(s')^2 + h^2][(h^2 + s^2)(h^2 + (s')^2) - \rho^2 s^2(s')^2]}\right.$$

$$\left. - \frac{-h^2[(1-\rho^2)(s')^2 + h^2]s^2 - h^4(h^2 + (s')^2)}{[(1-\rho^2)(s')^2 + h^2][(h^2 + s^2)(h^2 + (s')^2) - \rho^2 s^2(s')^2]}\right)$$

$$= -\frac{c^2}{2h^2}\left(1 - \frac{[(1-\rho^2)(s')^2 + h^2]\{[(1-\rho^2)(s')^2 + h^2]s^2 + 2h^2 ss'\rho - h^2 s^2\} - h^4[(1-\rho^2)(s')^2 + h^2]}{[(1-\rho^2)(s')^2 + h^2][(h^2 + s^2)(h^2 + (s')^2) - \rho^2 s^2(s')^2]}\right)$$

$$= -\frac{c^2}{2h^2}\left(1 - \frac{[(1-\rho^2)(s')^2 + h^2]\{s^2(s')^2 - \rho^2 s^2(s')^2 + 2h^2\rho ss' - h^4\}}{[(1-\rho^2)(s')^2 + h^2][(h^2 + s^2)(h^2 + (s')^2) - \rho^2 s^2(s')^2]}\right)$$

$$= -\frac{c^2}{2h^2}\left(1 - \frac{s^2(s')^2 - (h^2 - \rho ss')^2}{(h^2 + s^2)(h^2 + (s')^2) - \rho^2 s^2(s')^2}\right).$$

Combining the former results, we obtain

$$\mathbb{E}_{x\sim\mathcal{N}(0,1),z\sim\mathcal{N}(0,1-\rho^2)}\left[B(sx)B(s'(\rho x + z))\right]$$

$$= \frac{h^2}{\sqrt{(h^2 + s^2)(h^2 + (s')^2) - \rho^2 s^2(s')^2}}\exp\left(-\frac{c^2}{2h^2}\left(1 - \frac{s^2(s')^2 - (h^2 - \rho ss')^2}{(h^2 + s^2)(h^2 + (s')^2) - \rho^2 s^2(s')^2}\right)\right)$$

$$= \frac{h^2}{\sqrt{(h^2 + s^2)(h^2 + (s')^2) - \rho^2 s^2(s')^2}}\exp\left(-\frac{c^2}{2}\cdot\frac{(h^2 + s^2) + (h^2 + (s')^2) - 2\rho s(s')}{(h^2 + s^2)(h^2 + (s')^2) - \rho^2 s^2(s')^2}\right).$$

Then using the relations

$$s = \|x\|_2, \quad s' = \|x'\|_2, \quad \rho ss' = \langle x, x'\rangle,$$

we obtain

$$\mathbb{E}_{x\sim\mathcal{N}(0,1),z\sim\mathcal{N}(0,1-\rho^2)}\left[B(sx)B(s'(\rho x + z))\right]$$

$$= \frac{h^2}{\sqrt{(h^2 + \|x\|^2)(h^2 + \|x'\|^2) - \langle x, x'\rangle^2}}\exp\left(-\frac{c^2}{2}\cdot\frac{(h^2 + \|x\|^2) + (h^2 + \|x'\|^2) - 2\langle x, x'\rangle}{(h^2 + \|x\|^2)(h^2 + \|x'\|^2) - \langle x, x'\rangle^2}\right).$$

$\square$

### B.2 PROOF OF THEOREM 3.2

**Proof.**

**Step 1. Transform $K(r)$.** To obtain a uniform expression regardless of $h$, we transform $K(r)$ in the following manner.

$$K(r) = \frac{h^2}{\sqrt{(1+h^2)^2 - r^2}}\exp\left(-\frac{c^2}{1 + h^2 + r}\right)$$

$$= \frac{h^2}{1 + h^2}\frac{1}{\sqrt{1 - \left(\frac{r}{1+h^2}\right)^2}}\exp\left(-\frac{\frac{c^2}{1+h^2}}{1 + \frac{r}{1+h^2}}\right).$$

Let $p = \frac{c^2}{1+h^2} \in [0, +\infty)$, $u = \frac{r}{1+h^2} \in [-\frac{1}{1+h^2}, \frac{1}{1+h^2}] \subsetneq (-1, 1)$, and

$$f(u) := \frac{1}{\sqrt{1-u^2}} \exp\left(-\frac{p}{1+u}\right).$$

Then

$$K(r) = \frac{h^2}{1+h^2} f(u).$$

Hence we only need to consider the Taylor expansion of $f(u)$ where $u \in (-1, 1)$.

$$f(u) = \sum_{n=0}^{\infty} \frac{f^{(n)}(0)}{n!} u^n.$$

**Step 2. Deriving the recurrence relation of $f^{(n)}(0)$.** Solving the Taylor coefficients of $f(u)$ at $u = 0$ is highly technical. For starters, we derive the recurrence formula. For notational convenience, let $y = f(u)$.

From the definition of $y$, we have the equality

$$y\sqrt{1-u^2} = \exp\left(-\frac{p}{1+u}\right).$$

Taking derivatives on both sides, we have

$$y'\sqrt{1-u^2} - \frac{uy}{\sqrt{1-u^2}} = \frac{p}{(1+u)^2}\exp\left(-\frac{p}{1+u}\right)$$
$$= \frac{p\sqrt{1-u^2}}{(1+u)^2}y.$$

Multiplying $\frac{(u-1)^2(u+1)^2}{\sqrt{1-u^2}}$ on both sides, we have

$$(u^2-1)^2 y' + u(u^2-1)y = p(u-1)^2 y.$$

Eliminating the factor $(u-1)$ and expanding the polynomials lead to

$$(u^3 + u^2 - u - 1)y' + (u^2 + u)y = p(u-1)y.$$

Taking $n$-th derivatives on both sides and applying the Leibniz rule, we have

$$
\begin{array}{lll}
y^{(n+1)}(u^3 + u^2 - u - 1) & +y^{(n)}(u^2 + u) & \\
+ny^{(n)}(3u^2 + 2u - 1) & +ny^{(n-1)}(2u+1) = & y^{(n)}p(u-1) \\
+\frac{n(n-1)}{2}y^{(n-1)}(6u+2) & +\frac{n(n-1)}{2}y^{(n-2)}\cdot 2 & +ny^{(n-1)}p. \\
+\frac{n(n-1)(n-2)}{6}y^{(n-2)}\cdot 6 & &
\end{array}
$$

Let $u = 0$, and let $y^{(n)} = y^{(n)}(0)$ in the statements hereafter, we have that

$$-y^{(n+1)} - ny^{(n)} + n^2 y^{(n-1)} + n(n-1)^2 y^{(n-2)} = -py^{(n)} + npy^{(n-1)}.$$

Finally, we have the recurrence formula.

$$y^{(n+1)} = (p-n)y^{(n)} - n(p-n)y^{(n-1)} + n(n-1)^2 y^{(n-2)}. \tag{B.1}$$

To solve $\{y^{(n)}(0)\}_{n=0}^{\infty}$ from the recurrence relation, we also need to obtain $y(0), y'(0), y''(0)$ by hand. A simple calculation shows that

$$f(u) = \frac{1}{\sqrt{1-u^2}} \exp\left(-\frac{p}{1+u}\right)$$

$$f'(u) = \left(\frac{u}{\sqrt{(1-u^2)^3}} + \frac{1}{\sqrt{1-u^2}} \cdot \frac{p}{(1+u)^2}\right) \exp\left(-\frac{p}{1+u}\right)$$

$$f''(u) = \left(\frac{1}{\sqrt{(1-u^2)^3}} + \frac{3u^2}{\sqrt{(1-u^2)^5}}\right.$$

$$+ \frac{u}{\sqrt{(1-u^2)^3}} \cdot \frac{p}{(1+u)^2} + \frac{1}{\sqrt{1-u^2}} \cdot \frac{-2p}{(1+u)^3} +$$

$$\left.\left(\frac{u}{\sqrt{(1-u^2)^3}} + \frac{1}{\sqrt{1-u^2}} \cdot \frac{p}{(1+u)^2}\right) \frac{p}{(1+u)^2}\right) \exp\left(-\frac{p}{1+u}\right).$$

Hence, we obtain

$$y^{(0)} = e^{-p},$$
$$y^{(1)} = pe^{-p},$$
$$y^{(2)} = (p-1)^2 e^{-p}.$$

Solving $\{y^{(n)}\}_{n=0}^{\infty}$ remains to be difficult. To simplify the problem, we try to make some observations on the properties of $y^{(n)}$. We supplement $y^{(n)}$ till the first 8 terms.

$y^{(3)} = (p^3 - 6p^2 + 9p)e^{-p},$

$y^{(4)} = (p^4 - 12p^3 + 42p^2 - 36p + 9)e^{-p},$

$y^{(5)} = (p^5 - 20p^4 + 130p^3 - 300p^2 + 225p)e^{-p},$

$y^{(6)} = (p^6 - 30p^5 + 315p^4 - 1380p^3 + 2475p^2 - 1350p + 225)e^{-p},$

$y^{(7)} = (p^7 - 42p^6 + 651p^5 - 4620p^4 + 15435p^3 - 22050p^2 + 11025p)e^{-p},$

$y^{(8)} = (p^8 - 56p^7 + 1204p^6 - 12600p^5 + 67830p^4 - 182280p^3 + 220500p^2 - 88200p + 11025)e^{-p}.$

A further observation shows that

$$y^{(3)} = p(p-3)^2 e^{-p},$$
$$y^{(4)} = (p^2 - 6p + 3)^2 e^{-p},$$
$$y^{(5)} = p(p^2 - 10p + 15)^2 e^{-p},$$
$$y^{(6)} = (p^3 - 15p^2 + 45p - 15)^2 e^{-p},$$
$$y^{(7)} = p(p^3 - 21p^2 + 105p - 105)^2 e^{-p},$$
$$y^{(8)} = (p^4 - 28p^3 + 210p^2 - 420p + 105)^2 e^{-p}.$$

To conclude, we have the following observations.

1. $y^{(n)} = e^{-p} R_n(p)$, where $R_n(p)$ is a polynomial of degree $n$.
2. For $n = 2k$, $R_n(p) = P_k^2(p)$, where $P_k(p)$ is a polynomial of degree $k$.
3. For $n = 2k + 1$, $R_n(p) = p \cdot Q_k^2(p)$, where $Q_k(p)$ is a polynomial of degree $k$.

The correctness of the first observation is easily proved by induction. In the next step, we give a formal proof of the correctness of the second and third observations.

**Step 3. Formal proof of the general term formula of the Taylor coefficients.** The intuition of the proof is to directly derive the general term formula of $\{P_k\}$ and $\{Q_k\}$ from observations. Note that the observations are non-trivial.

We claim that

$$P_k(x) = \sum_{i=0}^{k} (-1)^{k-i} \frac{(2k-1)!!}{(2i-1)!!} \cdot \binom{k}{i} x^i, \tag{B.2}$$

and

$$Q_k(x) = \sum_{i=0}^{k} (-1)^{k-i} \frac{(2k+1)!!}{(2i+1)!!} \cdot \binom{k}{i} x^i, \tag{B.3}$$

where $(-1)!! := 1$, and $\binom{0}{0} := 1$ in the above expressions, and

$$R_{2k}(x) = P_k^2(x), \tag{B.4}$$

$$R_{2k+1}(x) = xQ_k^2(x). \tag{B.5}$$

We aim to prove the above four equalities true for all $k \in \mathbb{N}$ by induction.

First of all, it is easy to verify that the first three terms conform with the above expressions, where

$$P_0(x) = 1,$$
$$Q_0(x) = 1,$$
$$P_1(x) = x - 1,$$

and

$$y^{(0)} = P_0^2(p)e^{-p},$$
$$y^{(1)} = pQ_0^2(p)e^{-p},$$
$$y^{(2)} = P_1^2(p)e^{-p}.$$

For $n = 2k + 1$, where $k \geq 1$, suppose that Eq. (B.2) and Eq. (B.4) hold for all $i \leq k$ and Eq. (B.3) and Eq. (B.5) hold for all $i \leq k - 1$. We need to prove that Eq. (B.3) and Eq. (B.5) also hold for $i = k$. By Eq. (B.1), we only need to prove

$$xQ_k^2 = (x - 2k)P_k^2 - 2k(x - 2k)xQ_{k-1}^2 + 2k(2k-1)^2 P_{k-1}^2. \tag{B.6}$$

For $n = 2k$, where $k \geq 2$, suppose that Eq. (B.2) and Eq. (B.4) hold for all $i \leq k - 1$ and Eq. (B.3) and Eq. (B.5) hold for all $i \leq k - 1$. We need to prove that Eq. (B.2) and Eq. (B.4) also hold for $i = k$. By Eq. (B.1), we only need to prove

$$P_k^2 = (x - (2k-1))xQ_{k-1}^2 - (2k-1)(x - (2k-1))P_{k-1}^2 + (2k-1)(2k-2)^2 xQ_{k-2}^2. \tag{B.7}$$

For notational simplicity, we set for $i \in [k]$,

$$a_i^k = (-1)^{k-i} \frac{(2k-1)!!}{(2i-1)!!} \cdot \binom{k}{i}, \quad b_i^k = (-1)^{k-i} \frac{(2k+1)!!}{(2i+1)!!} \cdot \binom{k}{i}.$$

For $i \in [2k]$,

$$A_i^k = \sum_{j=0 \vee i-k}^{i \wedge k} a_j^k a_{i-j}^k, \quad B_i^k = \sum_{j=0 \vee i-k}^{i \wedge k} b_j^k b_{i-j}^k.$$

The polynomials are written as

$$P_k(x) = \sum_{i=0}^{k} a_i^k x^i, \quad Q_k(x) = \sum_{i=0}^{k} b_i^k x^i.$$

$$(P_k(x))^2 = \sum_{i=0}^{2k} A_i^k x^i, \quad (Q_k(x))^2 = \sum_{i=0}^{2k} B_i^k x^i.$$

**Proof of Eq. (B.6).**    Now consider the right-hand side of Eq. (B.6).

$$\text{RHS} = (x - 2k) \sum_{i=0}^{2k} A_i^k x^i - 2kx(x - 2k) \sum_{i=0}^{2k-2} B_i^{k-1} x^i + 2k(2k-1)^2 \sum_{i=0}^{2k-2} A_i^{k-1} x^i$$

$$= \sum_{i=0}^{2k} A_i^k x^{i+1} + \sum_{i=0}^{2k} (-2k) A_i^k x^i$$

$$+ \sum_{i=0}^{2k-2} (-2k) B_i^{k-1} x^{i+2} + \sum_{i=0}^{2k-2} (2k)^2 B_i^{k-1} x^{i+1}$$

$$+ \sum_{i=0}^{2k-2} 2k(2k-1)^2 A_i^{k-1} x^i$$

$$= \sum_{i=1}^{2k+1} A_{i-1}^k x^i + \sum_{i=0}^{2k} (-2k) A_i^k x^i$$

$$+ \sum_{i=2}^{2k} (-2k) B_{i-2}^{k-1} x^i + \sum_{i=1}^{2k-1} (2k)^2 B_{i-1}^{k-1} x^i$$

$$+ \sum_{i=0}^{2k-2} 2k(2k-1)^2 A_i^{k-1} x^i$$

$$= \sum_{i=2}^{2k-2} (A_{i-1}^k - 2k A_i^k - 2k B_{i-2}^{k-1} + (2k)^2 B_{i-1}^{k-1} + 2k(2k-1)^2 A_i^{k-1}) x^i$$

$$+ (A_{2k-2}^k - 2k A_{2k-1}^k - 2k B_{2k-3}^{k-1} + (2k)^2 B_{2k-2}^{k-1}) x^{2k-1}$$

$$+ (A_{2k-1}^k - 2k A_{2k}^k - 2k B_{2k-2}^{k-1}) x^{2k} + A_{2k}^k x^{2k+1}$$

$$+ (A_0^k - 2k A_1^k + (2k)^2 B_0^{k-1} + 2k(2k-1)^2 A_1^{k-1}) x$$

$$+ (2k(2k-1)^2 A_0^{k-1} - 2k A_0^k).$$

For the constant term, the general term formula is

$$A_0^k = (a_0^k)^2 = ((2k-1)!!)^2.$$

Hence,

$$2k(2k-1)^2 A_0^{k-1} - 2k A_0^k = 2k[(2k-1)^2 \cdot ((2k-3)!!)^2 - ((2k-1)!!)^2] = 0.$$

Plug the result into the right-hand side, we obatin

$$\text{RHS} = x \left\{ \sum_{i=2}^{2k-2} (A_{i-1}^k - 2k A_i^k - 2k B_{i-2}^{k-1} + (2k)^2 B_{i-1}^{k-1} + 2k(2k-1)^2 A_i^{k-1}) x^{i-1} \right.$$

$$+ (A_{2k-2}^k - 2k A_{2k-1}^k - 2k B_{2k-3}^{k-1} + (2k)^2 B_{2k-2}^{k-1}) x^{2k-2}$$

$$+ (A_{2k-1}^k - 2k A_{2k}^k - 2k B_{2k-2}^{k-1}) x^{2k-1} + A_{2k}^k x^{2k}$$

$$\left. + (A_0^k - 2k A_1^k + (2k)^2 B_0^{k-1} + 2k(2k-1)^2 A_1^{k-1}) \right\}$$

$$= x \left\{ \sum_{i=1}^{2k-3} (A_i^k - 2k A_{i+1}^k - 2k B_{i-1}^{k-1} + (2k)^2 B_i^{k-1} + 2k(2k-1)^2 A_{i+1}^{k-1}) x^i \right.$$

$$+ (A_{2k-2}^k - 2k A_{2k-1}^k - 2k B_{2k-3}^{k-1} + (2k)^2 B_{2k-2}^{k-1}) x^{2k-2}$$

$$+ (A_{2k-1}^k - 2k A_{2k}^k - 2k B_{2k-2}^{k-1}) x^{2k-1} + A_{2k}^k x^{2k}$$

$$\left. + (A_0^k - 2k A_1^k + (2k)^2 B_0^{k-1} + 2k(2k-1)^2 A_1^{k-1}) \right\}.$$

We then verify the coefficients are equal to those of $xQ_k^2(x) = x\left(\sum_{i=0}^{2k} B_i^k x^i\right)$.

For $i = 2k$,

$$A_{2k}^k = (a_k^k)^2 = 1^2 = (b_k^k)^2 = B_{2k}^k.$$

For $i = 2k - 1$,

$$
\begin{aligned}
&A_{2k-1}^k - 2kA_{2k}^k - 2kB_{2k-2}^{k-1} \\
=& 2a_{k-1}^k a_k^k - 2k(a_k^k)^2 - 2k(b_{k-1}^{k-1})^2 \\
=& -2(2k-1)k - 2k - 2k \\
=& -2k(2k+1) = B_{2k-1}^k.
\end{aligned}
$$

For $i = 2k - 2$,

$$
\begin{aligned}
&A_{2k-2}^k - 2kA_{2k-1}^k - 2kB_{2k-3}^{k-1} + (2k)^2 B_{2k-2}^{k-1} \\
=& (a_{k-1}^k)^2 + 2a_{k-2}^k a_k^k - 2k \cdot 2a_{k-1}^k a_k^k - 2k \cdot 2b_{k-2}^{k-1} b_{k-1}^{k-1} + (2k)^2 (b_{k-1}^{k-1})^2 \\
=& ((2k-1)\cdot k)^2 + 2\cdot(2k-1)(2k-3)\frac{k(k-1)}{2} + 2k\cdot 2(2k-1)k \\
&+ 2k\cdot 2(2k-1)(k-1) + (2k)^2 \\
=& ((2k+1)k)^2 + 2(2k+1)(2k-1)\frac{k(k-1)}{2} = B_{2k-2}^k.
\end{aligned}
$$

For $i = 0$,

$$
\begin{aligned}
&A_0^k - 2kA_1^k + (2k)^2 B_0^{k-1} + 2k(2k-1)^2 A_1^{k-1} \\
=& ((2k-1)!!)^2 - 2k(-2(2k-1)!! \cdot (2k-1)!! \cdot k) \\
&+ (2k)^2((2k-1)!!)^2 + 2k(2k-1)^2(-2(2k-3)!! \cdot (2k-3)!! \cdot (k-1)) \\
=& ((2k-1)!!)^2 + 2(2k)^2((2k-1)!!)^2 - (2k)(2k-2)((2k-1)!!)^2 \\
=& (2k+1)^2((2k-1)!!)^2 = ((2k+1)!!)^2 = B_0^k.
\end{aligned}
$$

For $1 \le i \le 2k - 3$, we need to show that

$$A_i^k - 2kA_{i+1}^k - 2kB_{i-1}^{k-1} + (2k)^2 B_i^{k-1} + 2k(2k-1)^2 A_{i+1}^{k-1} = B_i^k.$$

For starters, we have for the right-hand side that

$$
\begin{aligned}
(-1)^i B_i^k &= \sum_{j=0\vee i-k}^{i\wedge k} \frac{(2k+1)!!}{(2j+1)!!}\binom{k}{j}\frac{(2k+1)!!}{(2(i-j)+1)!!}\binom{k}{i-j} \\
&= (2k+1)^2 \sum_{j=0\vee i-k}^{i\wedge k} \frac{(2k-1)!!}{(2j+1)!!}\binom{k}{j}\frac{(2k-1)!!}{(2(i-j)+1)!!}\binom{k}{i-j}.
\end{aligned}
$$

For the left-hand side, we have

$$(-1)^i(A_i^k - 2kA_{i+1}^k - 2kB_{i-1}^{k-1} + (2k)^2 B_i^{k-1} + 2k(2k-1)^2 A_{i+1}^{k-1})$$

$$= \sum_{j=0\vee i-k}^{i\wedge k} \frac{(2k-1)!!}{(2j-1)!!}\binom{k}{j}\frac{(2k-1)!!}{(2(i-j)-1)!!}\binom{k}{i-j}$$

$$+ 2k\sum_{j=0\vee i+1-k}^{i+1\wedge k} \frac{(2k-1)!!}{(2j-1)!!}\binom{k}{j}\frac{(2k-1)!!}{(2(i+1-j)-1)!!}\binom{k}{i+1-j}$$

$$+ 2k\sum_{j=0\vee i-k}^{i-1\wedge k-1} \frac{(2k-1)!!}{(2j+1)!!}\binom{k-1}{j}\frac{(2k-1)!!}{(2(i-1-j)+1)!!}\binom{k-1}{i-1-j}$$

$$+ (2k)^2\sum_{j=0\vee i-k+1}^{i\wedge k-1} \frac{(2k-1)!!}{(2j+1)!!}\binom{k-1}{j}\frac{(2k-1)!!}{(2(i-j)+1)!!}\binom{k-1}{i-j}$$

$$- 2k(2k-1)^2\sum_{j=0\vee i-k+2}^{i+1\wedge k-1} \frac{(2k-3)!!}{(2j-1)!!}\binom{k-1}{j}\frac{(2k-3)!!}{(2(i+1-j)-1)!!}\binom{k-1}{i+1-j}$$

$$= \sum_{j=0\vee i-k}^{i\wedge k} \frac{(2k-1)!!}{(2j-1)!!}\binom{k}{j}\frac{(2k-1)!!}{(2(i-j)-1)!!}\binom{k}{i-j}$$

$$+ 2k\sum_{j=0\vee i+1-k}^{i+1\wedge k} \frac{(2k-1)!!}{(2j-1)!!}\binom{k}{j}\frac{(2k-1)!!}{(2(i-j)+1)!!}\binom{k}{i+1-j}$$

$$+ 2k\sum_{j=0\vee i-k}^{i-1\wedge k-1} \frac{(2k-1)!!}{(2j+1)!!}\binom{k-1}{j}\frac{(2k-1)!!}{(2(i-j)-1)!!}\binom{k-1}{i-1-j}$$

$$+ (2k)^2\sum_{j=0\vee i-k+1}^{i\wedge k-1} \frac{(2k-1)!!}{(2j+1)!!}\binom{k-1}{j}\frac{(2k-1)!!}{(2(i-j)+1)!!}\binom{k-1}{i-j}$$

$$- 2k\sum_{j=0\vee i-k+2}^{i+1\wedge k-1} \frac{(2k-1)!!}{(2j-1)!!}\binom{k-1}{j}\frac{(2k-1)!!}{(2(i-j)+1)!!}\binom{k-1}{i+1-j}$$

$$= \sum_{j=0\vee i-k}^{i\wedge k} [(2j+1)(2(i-j)+1)]\frac{(2k-1)!!}{(2j+1)!!}\binom{k}{j}\frac{(2k-1)!!}{(2(i-j)+1)!!}\binom{k}{i-j}$$

$$+ 2k\sum_{j=0\vee i+1-k}^{i+1\wedge k} \frac{(2k-1)!!}{(2j-1)!!}\binom{k}{j}\frac{(2k-1)!!}{(2(i-j)+1)!!}\binom{k}{i+1-j}$$

$$+ 2k\sum_{j=0\vee i-k}^{i-1\wedge k-1} \frac{(2k-1)!!}{(2j+1)!!}\binom{k-1}{j}\frac{(2k-1)!!}{(2(i-j)-1)!!}\binom{k-1}{i-1-j}$$

$$+ \sum_{j=0\vee i-k+1}^{i\wedge k-1} [2(k-j)2(k-(i-j))]\frac{(2k-1)!!}{(2j+1)!!}\binom{k}{j}\frac{(2k-1)!!}{(2(i-j)+1)!!}\binom{k}{i-j}$$

$$- 2k\sum_{j=0\vee i-k+2}^{i+1\wedge k-1} \frac{(2k-1)!!}{(2j-1)!!}\binom{k-1}{j}\frac{(2k-1)!!}{(2(i-j)+1)!!}\binom{k-1}{i+1-j}.$$

For the second, third and fifth terms, we have

$$
2k \sum_{j=0\vee i+1-k}^{i+1\wedge k} \frac{(2k-1)!!}{(2j-1)!!} \binom{k}{j} \frac{(2k-1)!!}{(2(i-j)+1)!!} \binom{k}{i+1-j}
$$

$$
+ 2k \sum_{j=0\vee i-k}^{i-1\wedge k-1} \frac{(2k-1)!!}{(2j+1)!!} \binom{k-1}{j} \frac{(2k-1)!!}{(2(i-j)-1)!!} \binom{k-1}{i-1-j}
$$

$$
- 2k \sum_{j=0\vee i-k+2}^{i+1\wedge k-1} \frac{(2k-1)!!}{(2j-1)!!} \binom{k-1}{j} \frac{(2k-1)!!}{(2(i-j)+1)!!} \binom{k-1}{i+1-j}
$$

$$
=2k \sum_{j=0\vee i+1-k}^{i+1\wedge k} \frac{(2k-1)!!}{(2j-1)!!} \binom{k}{j} \frac{(2k-1)!!}{(2(i-j)+1)!!} \binom{k}{i+1-j}
$$

$$
+ 2k \sum_{j=0\vee i+1-k}^{i\wedge k-1} \frac{(2k-1)!!}{(2j+1)!!} \binom{k-1}{j} \frac{(2k-1)!!}{(2(i-j)-1)!!} \binom{k}{i-j}
$$

$$
- 2k \sum_{j=0\vee i+1-k}^{i\wedge k-1} \frac{(2k-1)!!}{(2j+1)!!} \binom{k-1}{j} \frac{(2k-1)!!}{(2(i-j)-1)!!} \binom{k-1}{i-j}
$$

$$
- 2k \sum_{j=0\vee i-k+2}^{i+1\wedge k-1} \frac{(2k-1)!!}{(2j-1)!!} \binom{k-1}{j} \frac{(2k-1)!!}{(2(i-j)+1)!!} \binom{k-1}{i+1-j}
$$

$$
=2k \sum_{j=0\vee i+1-k}^{i+1\wedge k} \frac{(2k-1)!!}{(2j-1)!!} \frac{k!}{j!(k-j)!} \frac{(2k-1)!!}{(2(i-j)+1)!!} \frac{k!}{(i-j+1)!(k-i+j-1)!}
$$

$$
+ 2k \sum_{j=0\vee i+1-k}^{i\wedge k-1} \frac{(2k-1)!!}{(2j+1)!!} \frac{(k-1)!}{j!(k-j-1)!} \frac{(2k-1)!!}{(2(i-j)-1)!!} \frac{k!}{(i-j)!(k-i+j)!}
$$

$$
- 2k \sum_{j=0\vee i+1-k}^{i\wedge k-1} \frac{(2k-1)!!}{(2j+1)!!} \frac{(k-1)!}{j!(k-j-1)!} \frac{(2k-1)!!}{(2(i-j)-1)!!} \frac{(k-1)!}{(i-j)!(k-i+j-1)!}
$$

$$
- 2k \sum_{j=0\vee i-k+2}^{i+1\wedge k-1} \frac{(2k-1)!!}{(2j-1)!!} \frac{(k-1)!}{j!(k-j-1)!} \frac{(2k-1)!!}{(2(i-j)+1)!!} \frac{(k-1)!}{(i-j+1)!(k-i+j-2)!}
$$

$$
=2k \sum_{j=0\vee i+1-k}^{i+1\wedge k} \frac{(2k-1)!!}{(2j-1)!!} \frac{k!}{j!(k-j)!} \frac{(2k-1)!!}{(2(i-j)+1)!!} \frac{k!}{(i-j+1)!(k-i+j-1)!}
$$

$$
+ \sum_{j=0\vee i+1-k}^{i\wedge k-1} [2(k-j)(2(i-j)+1)]\frac{(2k-1)!!}{(2j+1)!!} \frac{k!}{j!(k-j)!} \frac{(2k-1)!!}{(2(i-j)+1)!!} \frac{k!}{(i-j)!(k-i+j)!}
$$

$$
- 2k \sum_{j=0\vee i+1-k}^{i\wedge k-1} \frac{(2k-1)!!}{(2j+1)!!} \frac{(k-1)!}{j!(k-j-1)!} \frac{(2k-1)!!}{(2(i-j)-1)!!} \frac{(k-1)!}{(i-j)!(k-i+j-1)!}
$$

$$
- 2k \sum_{j=0\vee i-k+2}^{i+1\wedge k-1} \frac{(2k-1)!!}{(2j-1)!!} \frac{(k-1)!}{j!(k-j-1)!} \frac{(2k-1)!!}{(2(i-j)+1)!!} \frac{(k-1)!}{(i-j+1)!(k-i+j-2)!},
$$

where in the first equality, we use the relation

$$
\binom{k}{i-j} - \binom{k-1}{i-j} = \binom{k-1}{i-j-1}.
$$

For the first, third and fourth terms of the former expression, we have

$$2k \sum_{j=0 \vee i+1-k}^{i+1 \wedge k} \frac{(2k-1)!!}{(2j-1)!!} \frac{k!}{j!(k-j)!} \frac{(2k-1)!!}{(2(i-j)+1)!!} \frac{k!}{(i-j+1)!(k-i+j-1)!}$$

$$- 2k \sum_{j=0 \vee i+1-k}^{i \wedge k-1} \frac{(2k-1)!!}{(2j+1)!!} \frac{(k-1)!}{j!(k-j-1)!} \frac{(2k-1)!!}{(2(i-j)-1)!!} \frac{(k-1)!}{(i-j)!(k-i+j-1)!}$$

$$- 2k \sum_{j=0 \vee i-k+2}^{i+1 \wedge k-1} \frac{(2k-1)!!}{(2j-1)!!} \frac{(k-1)!}{j!(k-j-1)!} \frac{(2k-1)!!}{(2(i-j)+1)!!} \frac{(k-1)!}{(i-j+1)!(k-i+j-2)!}$$

$$=2k \sum_{j=0 \vee i+1-k}^{i+1 \wedge k} \frac{(2k-1)!!}{(2j-1)!!} \frac{k!}{j!(k-j)!} \frac{(2k-1)!!}{(2(i-j)+1)!!} \frac{k!}{(i-j+1)!(k-i+j-1)!}$$

$$- 2k \sum_{j=0 \vee i+1-k}^{i \wedge k-1} \frac{(2k-1)!!}{(2j+1)!!} \frac{(k-1)!}{j!(k-j-1)!} \frac{(2k-1)!!}{(2(i-j)-1)!!} \frac{(k-1)!}{(i-j)!(k-i+j-1)!}$$

$$- 2 \sum_{j=0 \vee i-k+2}^{i+1 \wedge k-1} (k-j) \frac{(2k-1)!!}{(2j-1)!!} \frac{k!}{j!(k-j)!} \frac{(2k-1)!!}{(2(i-j)+1)!!} \frac{(k-1)!}{(i-j+1)!(k-i+j-2)!}$$

$$=2k \sum_{j=0 \vee i+1-k}^{i+1 \wedge k} \frac{(2k-1)!!}{(2j-1)!!} \frac{k!}{j!(k-j)!} \frac{(2k-1)!!}{(2(i-j)+1)!!} \frac{k!}{(i-j+1)!(k-i+j-1)!}$$

$$- 2k \sum_{j=0 \vee i+1-k}^{i \wedge k-1} \frac{(2k-1)!!}{(2j+1)!!} \frac{(k-1)!}{j!(k-j-1)!} \frac{(2k-1)!!}{(2(i-j)-1)!!} \frac{(k-1)!}{(i-j)!(k-i+j-1)!}$$

$$- 2k \sum_{j=0 \vee i-k+2}^{i+1 \wedge k-1} \frac{(2k-1)!!}{(2j-1)!!} \frac{k!}{j!(k-j)!} \frac{(2k-1)!!}{(2(i-j)+1)!!} \frac{(k-1)!}{(i-j+1)!(k-i+j-2)!}$$

$$+ \sum_{j=0 \vee i-k+2}^{i+1 \wedge k-1} (2j) \frac{(2k-1)!!}{(2j-1)!!} \frac{k!}{j!(k-j)!} \frac{(2k-1)!!}{(2(i-j)+1)!!} \frac{(k-1)!}{(i-j+1)!(k-i+j-2)!}$$

$$=2k \sum_{j=0 \vee i-k}^{i \wedge k} \frac{(2k-1)!!}{(2j-1)!!} \frac{k!}{j!(k-j)!} \frac{(2k-1)!!}{(2(i-j)+1)!!} \frac{(k-1)!}{(i-j)!(k-i+j-1)!}$$

$$- 2k \sum_{j=0 \vee i+1-k}^{i \wedge k-1} \frac{(2k-1)!!}{(2j+1)!!} \frac{(k-1)!}{j!(k-j-1)!} \frac{(2k-1)!!}{(2(i-j)-1)!!} \frac{(k-1)!}{(i-j)!(k-i+j-1)!}$$

$$+ \sum_{j=0 \vee i-k+2}^{i+1 \wedge k-1} (2j) \frac{(2k-1)!!}{(2j-1)!!} \frac{k!}{j!(k-j)!} \frac{(2k-1)!!}{(2(i-j)+1)!!} \frac{(k-1)!}{(i-j+1)!(k-i+j-2)!}$$

$$= \sum_{j=0 \vee i-k}^{i \wedge k} [(2j+1)2(k-i+j)] \frac{(2k-1)!!}{(2j+1)!!} \frac{k!}{j!(k-j)!} \frac{(2k-1)!!}{(2(i-j)+1)!!} \frac{(k-1)!}{(i-j)!(k-i+j-1)!}$$

$$- 2k \sum_{j=0 \vee i+1-k}^{i \wedge k-1} \frac{(2k-1)!!}{(2j+1)!!} \frac{(k-1)!}{j!(k-j-1)!} \frac{(2k-1)!!}{(2(i-j)-1)!!} \frac{(k-1)!}{(i-j)!(k-i+j-1)!}$$

$$+ \sum_{j=0 \vee i-k+2}^{i+1 \wedge k-1} (2j) \frac{(2k-1)!!}{(2j-1)!!} \frac{k!}{j!(k-j)!} \frac{(2k-1)!!}{(2(i-j)+1)!!} \frac{(k-1)!}{(i-j+1)!(k-i+j-2)!},$$

where in the third equality, we combine the first and third terms using the relation

$$\binom{k}{i-j+1} - \binom{k-1}{i-j+1} = \binom{k-1}{i-j}.$$

For the last two terms, we have

$$
-2k\sum_{j=0\vee i+1-k}^{i\wedge k-1}\frac{(2k-1)!!}{(2j+1)!!}\frac{(k-1)!}{j!(k-j-1)!}\frac{(2k-1)!!}{(2(i-j)-1)!!}\frac{(k-1)!}{(i-j)!(k-i+j-1)!}
$$

$$
+\sum_{j=0\vee i-k+2}^{i+1\wedge k-1}(2j)\frac{(2k-1)!!}{(2j-1)!!}\frac{k!}{j!(k-j)!}\frac{(2k-1)!!}{(2(i-j)+1)!!}\frac{(k-1)!}{(i-j+1)!(k-i+j-2)!}
$$

$$
=-2k\sum_{j=0\vee i+1-k}^{i\wedge k-1}\frac{(2k-1)!!}{(2j+1)!!}\frac{(k-1)!}{j!(k-j-1)!}\frac{(2k-1)!!}{(2(i-j)-1)!!}\frac{(k-1)!}{(i-j)!(k-i+j-1)!}
$$

$$
+\sum_{j=0\vee i-k+1}^{i\wedge k-1}(2j+2)\frac{(2k-1)!!}{(2j+1)!!}\frac{k!}{(j+1)!(k-j-1)!}\frac{(2k-1)!!}{(2(i-j)-1)!!}\frac{(k-1)!}{(i-j)!(k-i+j-1)!}
$$

$$
=-2\sum_{j=0\vee i+1-k}^{i\wedge k-1}\frac{(2k-1)!!}{(2j+1)!!}\frac{k!}{j!(k-j-1)!}\frac{(2k-1)!!}{(2(i-j)-1)!!}\frac{(k-1)!}{(i-j)!(k-i+j-1)!}
$$

$$
+2\sum_{j=0\vee i-k+1}^{i\wedge k-1}\frac{(2k-1)!!}{(2j+1)!!}\frac{k!}{j!(k-j-1)!}\frac{(2k-1)!!}{(2(i-j)-1)!!}\frac{(k-1)!}{(i-j)!(k-i+j-1)!}
$$

$$
=0.
$$

Combine the four parts illustrated above, we have that

$$
(-1)^i(A_i^k-2kA_{i+1}^k-2kB_{i-1}^{k-1}+(2k)^2B_i^{k-1}+2k(2k-1)^2A_{i+1}^{k-1})
$$

$$
=\sum_{j=0\vee i-k}^{i\wedge k}[(2j+1)(2(i-j)+1)+2(k-j)2(k-i+j)
$$

$$
+2(k-j)(2(i-j)+1)+(2j+1)2(k-i+j)+0]
$$

$$
\cdot\frac{(2k-1)!!}{(2j+1)!!}\binom{k}{j}\frac{(2k-1)!!}{(2(i-j)+1)!!}\binom{k}{i-j}
$$

$$
=\sum_{j=0\vee i-k}^{i\wedge k}(2k+1)^2\frac{(2k-1)!!}{(2j+1)!!}\binom{k}{j}\frac{(2k-1)!!}{(2(i-j)+1)!!}\binom{k}{i-j}
$$

$$
=\sum_{j=0\vee i-k}^{i\wedge k}\frac{(2k+1)!!}{(2j+1)!!}\binom{k}{j}\frac{(2k+1)!!}{(2(i-j)+1)!!}\binom{k}{i-j}
$$

$$
=(-1)^iB_i^k.
$$

Finally, we complete the proof of Eq. (B.6).

**Proof of Eq. (B.7).**   Consider the right-hand side of Eq. (B.7).

$$\text{RHS} =(x - (2k-1))x \sum_{i=0}^{2k-2} B_i^{k-1} x^i - (2k-1)(x - (2k-1)) \sum_{i=0}^{2k-2} A_i^{k-1} x^i + (2k-1)(2k-2)^2 x \sum_{i=0}^{2k-4} B_i^{k-2} x^i$$

$$= \sum_{i=0}^{2k-2} B_i^{k-1} x^{i+2} + \sum_{i=0}^{2k-2} (-(2k-1)) B_i^{k-1} x^{i+1}$$

$$+ \sum_{i=0}^{2k-2} (-(2k-1)) A_i^{k-1} x^{i+1} + \sum_{i=0}^{2k-2} (2k-1)^2 A_i^{k-1} x^i$$

$$+ \sum_{i=0}^{2k-4} (2k-1)(2k-2)^2 B_i^{k-2} x^{i+1}$$

$$= \sum_{i=2}^{2k} B_{i-2}^{k-1} x^i + \sum_{i=1}^{2k-1} (-(2k-1)) B_{i-1}^{k-1} x^i$$

$$+ \sum_{i=1}^{2k-1} (-(2k-1)) A_{i-1}^{k-1} x^i + \sum_{i=0}^{2k-2} (2k-1)^2 A_i^{k-1} x^i$$

$$+ \sum_{i=1}^{2k-3} (2k-1)(2k-2)^2 B_{i-1}^{k-2} x^i$$

$$= \sum_{i=2}^{2k-3} (B_{i-2}^{k-1} - (2k-1)B_{i-1}^{k-1} - (2k-1)A_{i-1}^{k-1} + (2k-1)^2 A_i^{k-1} + (2k-1)(2k-2)^2 B_{i-1}^{k-2}) x^i$$

$$+ (B_{2k-4}^{k-1} - (2k-1)B_{2k-3}^{k-1} - (2k-1)A_{2k-3}^{k-1} + (2k-1)^2 A_{2k-2}^{k-1}) x^{2k-2}$$

$$+ (B_{2k-3}^{k-1} - (2k-1)B_{2k-2}^{k-1} - (2k-1)A_{2k-2}^{k-1}) x^{2k-1} + B_{2k-2}^{k-1} x^{2k}$$

$$+ (-(2k-1)B_0^{k-1} - (2k-1)A_0^{k-1} + (2k-1)^2 A_1^{k-1} + (2k-1)(2k-2)^2 B_0^{k-2}) x$$

$$+ (2k-1)^2 A_0^{k-1}.$$

It suffices to prove that the coefficients of the above expression are equal to those of $P_k^2(x) = \sum_{i=0}^{2k} A_i^k x^i$.

For $i = 0, 1, 2k-2, 2k-1, 2k$, the verifications are trivial. We only need to show that for $2 \le i \le 2k-3$, it holds that

$$B_{i-2}^{k-1} - (2k-1)B_{i-1}^{k-1} - (2k-1)A_{i-1}^{k-1} + (2k-1)^2 A_i^{k-1} + (2k-1)(2k-2)^2 B_{i-1}^{k-2} = A_i^k.$$

Consider the right-hand side, we have

$$(-1)^i A_i^k = \sum_{j=0 \vee i-k}^{i \wedge k} \frac{(2k-1)!!}{(2j-1)!!} \binom{k}{j} \frac{(2k-1)!!}{(2(i-j)-1)!!} \binom{k}{i-j}.$$

For the left-hand side, we first have

$$(-1)^i(B_{i-2}^{k-1} - (2k-1)B_{i-1}^{k-1} - (2k-1)A_{i-1}^{k-1} + (2k-1)^2 A_i^{k-1} + (2k-1)(2k-2)^2 B_{i-1}^{k-2})$$

$$= \sum_{j=0\vee i-1-k}^{i-2\wedge k-1} \frac{(2k-1)!!}{(2j+1)!!}\binom{k-1}{j}\frac{(2k-1)!!}{(2(i-2-j)+1)!!}\binom{k-1}{i-2-j}$$

$$+ (2k-1)\sum_{j=0\vee i-k}^{i-1\wedge k-1} \frac{(2k-1)!!}{(2j+1)!!}\binom{k-1}{j}\frac{(2k-1)!!}{(2(i-1-j)+1)!!}\binom{k-1}{i-1-j}$$

$$+ (2k-1)\sum_{j=0\vee i-k}^{i-1\wedge k-1} \frac{(2k-3)!!}{(2j-1)!!}\binom{k-1}{j}\frac{(2k-3)!!}{(2(i-1-j)-1)!!}\binom{k-1}{i-1-j}$$

$$+ (2k-1)^2\sum_{j=0\vee i-k+1}^{i\wedge k-1} \frac{(2k-3)!!}{(2j-1)!!}\binom{k-1}{j}\frac{(2k-3)!!}{(2(i-j)-1)!!}\binom{k-1}{i-j}$$

$$- (2k-1)(2k-2)^2\sum_{j=0\vee i-k+1}^{i-1\wedge k-2} \frac{(2k-3)!!}{(2j+1)!!}\binom{k-2}{j}\frac{(2k-3)!!}{(2(i-1-j)+1)!!}\binom{k-2}{i-1-j}$$

$$= \sum_{j=0\vee i-1-k}^{i-2\wedge k-1} \frac{(2k-1)!!}{(2j+1)!!}\binom{k-1}{j}\frac{(2k-1)!!}{(2(i-j)-3)!!}\binom{k-1}{i-2-j}$$

$$+ (2k-1)\sum_{j=0\vee i-k}^{i-1\wedge k-1} \frac{(2k-1)!!}{(2j+1)!!}\binom{k-1}{j}\frac{(2k-1)!!}{(2(i-j)-1)!!}\binom{k-1}{i-1-j}$$

$$+ (2k-1)\sum_{j=0\vee i-k}^{i-1\wedge k-1} \frac{(2k-3)!!}{(2j-1)!!}\binom{k-1}{j}\frac{(2k-3)!!}{(2(i-j)-3)!!}\binom{k-1}{i-1-j}$$

$$+ (2k-1)^2\sum_{j=0\vee i-k+1}^{i\wedge k-1} \frac{(2k-3)!!}{(2j-1)!!}\binom{k-1}{j}\frac{(2k-3)!!}{(2(i-j)-1)!!}\binom{k-1}{i-j}$$

$$- (2k-1)(2k-2)^2\sum_{j=0\vee i-k+1}^{i-1\wedge k-2} \frac{(2k-3)!!}{(2j+1)!!}\binom{k-2}{j}\frac{(2k-3)!!}{(2(i-j)-1)!!}\binom{k-2}{i-1-j}$$

$$= \sum_{j=0\vee i-k}^{i-1\wedge k-1} \frac{(2k-1)!!}{(2j+1)!!}\binom{k-1}{j}\frac{(2k-1)!!}{(2(i-j)-3)!!}\binom{k}{i-1-j}$$

$$- \sum_{j=0\vee i-k}^{i-1\wedge k-1} \frac{(2k-1)!!}{(2j+1)!!}\binom{k-1}{j}\frac{(2k-1)!!}{(2(i-j)-3)!!}\binom{k-1}{i-1-j}$$

$$+ (2k-1)\sum_{j=0\vee i-k}^{i-1\wedge k-1} \frac{(2k-1)!!}{(2j+1)!!}\binom{k-1}{j}\frac{(2k-1)!!}{(2(i-j)-1)!!}\binom{k-1}{i-1-j}$$

$$+ (2k-1)\sum_{j=0\vee i-k}^{i-1\wedge k-1} \frac{(2k-3)!!}{(2j-1)!!}\binom{k-1}{j}\frac{(2k-3)!!}{(2(i-j)-3)!!}\binom{k-1}{i-1-j}$$

$$+ \sum_{j=0\vee i-k+1}^{i\wedge k-1} \frac{(2k-1)!!}{(2j-1)!!}\binom{k-1}{j}\frac{(2k-1)!!}{(2(i-j)-1)!!}\binom{k-1}{i-j}$$

$$- (2k-1)\sum_{j=0\vee i-k+1}^{i-1\wedge k-2} 2(k-1-j)2(k-i+j)\frac{(2k-3)!!}{(2j+1)!!}\binom{k-1}{j}\frac{(2k-3)!!}{(2(i-j)-1)!!}\binom{k-1}{i-1-j}.$$

To continue, we have

$$
\begin{aligned}
\text{LHS} =& \sum_{j=(0\vee i-k)+1}^{i\wedge k} \frac{(2k-1)!!}{(2j-1)!!}\binom{k-1}{j-1}\frac{(2k-1)!!}{(2(i-j)-1)!!}\binom{k}{i-j} \\
&+ \sum_{j=0\vee i-k+1}^{i\wedge k-1} \frac{(2k-1)!!}{(2j-1)!!}\binom{k-1}{j}\frac{(2k-1)!!}{(2(i-j)-1)!!}\binom{k-1}{i-j} \\
&- \sum_{j=0\vee i-k}^{i-1\wedge k-1} \frac{(2k-1)!!}{(2j+1)!!}\binom{k-1}{j}\frac{(2k-1)!!}{(2(i-j)-3)!!}\binom{k-1}{i-1-j} \\
&+ (2k-1)\sum_{j=0\vee i-k}^{i-1\wedge k-1} \frac{(2k-1)!!}{(2j+1)!!}\binom{k-1}{j}\frac{(2k-1)!!}{(2(i-j)-1)!!}\binom{k-1}{i-1-j} \\
&+ (2k-1)\sum_{j=0\vee i-k}^{i-1\wedge k-1} \frac{(2k-3)!!}{(2j-1)!!}\binom{k-1}{j}\frac{(2k-3)!!}{(2(i-j)-3)!!}\binom{k-1}{i-1-j} \\
&- (2k-1)\sum_{j=0\vee i-k+1}^{i-1\wedge k-2} 2(k-1-j)2(k-i+j)\frac{(2k-3)!!}{(2j+1)!!}\binom{k-1}{j}\frac{(2k-3)!!}{(2(i-j)-1)!!}\binom{k-1}{i-1-j} \\
=& \sum_{j=(0\vee i-k)+1}^{i\wedge k} \frac{(2k-1)!!}{(2j-1)!!}\binom{k-1}{j-1}\frac{(2k-1)!!}{(2(i-j)-1)!!}\binom{k}{i-j} \\
&+ \sum_{j=0\vee i-k+1}^{i\wedge k-1} \frac{(2k-1)!!}{(2j-1)!!}\binom{k-1}{j}\frac{(2k-1)!!}{(2(i-j)-1)!!}\binom{k-1}{i-j} \\
&- \sum_{j=0\vee i-k}^{i-1\wedge k-1} (2k-1)(2(i-j)-1)\frac{(2k-3)!!}{(2j+1)!!}\binom{k-1}{j}\frac{(2k-1)!!}{(2(i-j)-1)!!}\binom{k-1}{i-1-j} \\
&+ \sum_{j=0\vee i-k}^{i-1\wedge k-1} (2k-1)^2\frac{(2k-3)!!}{(2j+1)!!}\binom{k-1}{j}\frac{(2k-1)!!}{(2(i-j)-1)!!}\binom{k-1}{i-1-j} \\
&+ \sum_{j=0\vee i-k}^{i-1\wedge k-1} (2j+1)(2(i-j)-1)\frac{(2k-3)!!}{(2j+1)!!}\binom{k-1}{j}\frac{(2k-1)!!}{(2(i-j)-1)!!}\binom{k-1}{i-1-j} \\
&- \sum_{j=0\vee i-k+1}^{i-1\wedge k-2} (2k-2(j+1))(2k-2(i-j))\frac{(2k-3)!!}{(2j+1)!!}\binom{k-1}{j}\frac{(2k-1)!!}{(2(i-j)-1)!!}\binom{k-1}{i-1-j}.
\end{aligned}
$$

We consider combining the last four terms. Because

$$
\begin{aligned}
&-(2k-1)(2(i-j)-1)+(2k-1)^2 \\
&\quad +(2j+1)(2(i-j)-1)-(2k-2(j+1))(2k-2(i-j)) \\
=&-(2k-2j-2)(2(i-j)-1)+(2k-1)^2 \\
&\quad -(2k-2j-2)(2k-2(i-j)) \\
=&(2k-1)(2k-1-2k+2j+2) \\
=&(2k-1)(2j+1),
\end{aligned}
$$

we have

$$
\begin{aligned}
\text{LHS} =& \sum_{j=(0 \vee i-k)+1}^{i \wedge k} \frac{(2k-1)!!}{(2j-1)!!}\binom{k-1}{j-1}\frac{(2k-1)!!}{(2(i-j)-1)!!}\binom{k}{i-j} \\
&+ \sum_{j=0 \vee i-k+1}^{i \wedge k-1} \frac{(2k-1)!!}{(2j-1)!!}\binom{k-1}{j}\frac{(2k-1)!!}{(2(i-j)-1)!!}\binom{k-1}{i-j} \\
&+ \sum_{j=0 \vee i-k}^{i-1 \wedge k-1} (2k-1)(2j+1)\frac{(2k-3)!!}{(2j+1)!!}\binom{k-1}{j}\frac{(2k-1)!!}{(2(i-j)-1)!!}\binom{k-1}{i-1-j} \\
=& \sum_{j=(0 \vee i-k)+1}^{i \wedge k} \frac{(2k-1)!!}{(2j-1)!!}\binom{k-1}{j-1}\frac{(2k-1)!!}{(2(i-j)-1)!!}\binom{k}{i-j} \\
&+ \sum_{j=0 \vee i-k+1}^{i \wedge k-1} \frac{(2k-1)!!}{(2j-1)!!}\binom{k-1}{j}\frac{(2k-1)!!}{(2(i-j)-1)!!}\binom{k-1}{i-j} \\
&+ \sum_{j=0 \vee i-k}^{i-1 \wedge k-1} \frac{(2k-1)!!}{(2j-1)!!}\binom{k-1}{j}\frac{(2k-1)!!}{(2(i-j)-1)!!}\binom{k-1}{i-1-j} \\
=& \sum_{j=0 \vee i-k+1}^{i \wedge k-1} \frac{(2k-1)!!}{(2j-1)!!}\binom{k}{j}\frac{(2k-1)!!}{(2(i-j)-1)!!}\binom{k}{i-j} \\
=& (-1)^i A_i^k.
\end{aligned}
$$

Finally, we complete the proof of Eq. (B.7).

$\square$

## B.3 PROOF OF COROLLARY 3.3

**Proof.** The Taylor expansion of $K(r)$ is

$$
\begin{aligned}
K(r) =& e^{-p}\frac{h^2}{1+h^2}\sum_{n=0}^{\infty}\frac{R_n(p)}{n!(1+h^2)^n}r^n \\
=& \sum_{n=0}^{\infty}\frac{e^{-p}R_n(p)}{n!}\cdot\frac{h^2}{(1+h^2)^{n+1}}\langle x,x'\rangle^n \\
=& \sum_{n=0}^{\infty}\frac{e^{-p}R_n(p)}{n!}\cdot\frac{h^2}{(1+h^2)^{n+1}}\left\langle x^{\otimes n},x'^{\otimes n}\right\rangle \\
=& \sum_{n=0}^{\infty}\left\langle \frac{he^{-\frac{p}{2}}R_n^{\frac{1}{2}}(p)}{\sqrt{n!(1+h^2)^{n+1}}}x^{\otimes n},\frac{he^{-\frac{p}{2}}R_n^{\frac{1}{2}}(p)}{\sqrt{n!(1+h^2)^{n+1}}}x'^{\otimes n}\right\rangle.
\end{aligned} \tag{B.8}
$$

Hence the feature mapping with respect to kernel (3.4) is

$$
\phi(x)=\left(\frac{he^{-\frac{p}{2}}R_n^{\frac{1}{2}}(p)}{\sqrt{n!(1+h^2)^{n+1}}}x^{\otimes n}\right)_{n=0}^{\infty}.
$$

For any target function

$$
f(x)=\sum_{n=0}^{\infty}\langle F_n,x^{\otimes n}\rangle,
$$

where $F_n \in \mathbb{R}^{d^n}$, we have

$$f(x) = \sum_{n=0}^{\infty} \langle F_n, x^{\otimes n} \rangle$$

$$= \left\langle \frac{\sqrt{n!(1+h^2)^{n+1}}}{he^{-\frac{p}{2}} R_n^{\frac{1}{2}}(p)} F_n, (\phi(x))_n \right\rangle.$$

Hence, we have

$$\|f\|_{\mathcal{H}_K}^2 \leq \left\| \left( \frac{\sqrt{n!(1+h^2)^{n+1}}}{he^{-\frac{p}{2}} R_n^{\frac{1}{2}}(p)} F_n \right)_{n=0}^{\infty} \right\|_{\mathcal{H}}^2$$

$$= \sum_{n=0}^{\infty} \frac{n!(1+h^2)^{n+1}}{h^2 e^{-p} R_n(p)} \|F_n\|_{\mathrm{Fr}}^2$$

$$= \frac{e^p}{h^2} \sum_{n=0}^{\infty} \frac{n!(1+h^2)^{n+1}}{R_n(p)} \|F_n\|_{\mathrm{Fr}}^2,$$

where $\| \cdot \|_{\mathrm{Fr}}$ the the Frobenius norm.

Let

$$D(f) := \frac{e^p}{h^2} \sum_{n=0}^{\infty} \frac{n!(1+h^2)^{n+1}}{R_n(p)} \|F_n\|_{\mathrm{Fr}}^2.$$

By Proposition (2.1), we conclude that

$$\|f\|_{\mathcal{F}} = \|f\|_{\mathcal{H}_K} \leq \sqrt{D(f)}.$$

Furthermore, there exist $v : \mathcal{W} \to \mathbb{R}$ such that

$$f(x) = \mathbb{E}_{w \sim \mathcal{N}(0, I_d)} \left[ B(w^{\top} x) v(w) \right],$$

and

$$\|v\|_{\mathcal{H}_{\mathcal{W}}} \leq \sqrt{D(f)}.$$

$\square$

### B.4 PROOF OF THEOREM 3.4

**Proof.** Let $W = (w_1, w_2, ..., w_M)$ and $v_m = v(w_m)$. We already have $\varphi(x) := \mathbb{E}_{w \sim \mathcal{N}(0, I_d)} \left[ B(w^{\top} x) v(w) \right]$.

To obtain the desired result, we consider the concentration property of the random variable

$$\mathbb{E}_x |\hat{\varphi}(x) - \varphi(x)| = \mathbb{E}_x \left| \frac{1}{M} \sum_{m=1}^{M} B(w_m^{\top} x) v(w_m) - \varphi(x) \right|,$$

in which the randomness comes from $W$.

Naturally, we consider

$$\mathbb{E}_W \exp \left( \lambda^2 \left( \mathbb{E}_x \left| \frac{1}{M} \sum_{m=1}^{M} B(w_m^{\top} x) v(w_m) - \varphi(x) \right| \right)^2 \right)$$

$$\leq \mathbb{E}_W \exp \left( \lambda^2 \mathbb{E}_x \left( \frac{1}{M} \sum_{m=1}^{M} B(w_m^{\top} x) v(w_m) - \varphi(x) \right)^2 \right)$$

$$\leq \mathbb{E}_W \mathbb{E}_x \exp \left( \lambda^2 \left( \frac{1}{M} \sum_{m=1}^{M} B(w_m^{\top} x) v(w_m) - \varphi(x) \right)^2 \right) \tag{B.9}$$

$$= \mathbb{E}_x \mathbb{E}_W \exp \left( \lambda^2 \left( \frac{1}{M} \sum_{m=1}^{M} B(w_m^{\top} x) v(w_m) - \varphi(x) \right)^2 \right),$$

where we used Jensen's inequality twice.

Next, we prove that $B(w_m^\top x)v(w_m) - \varphi(x)$ are sub-gaussian random variables for every $w_m \sim \mathcal{N}(0, I_d)$ and every $x \in \mathbb{R}$. In addition, they have a uniform sub-gaussian norm.

To start with, for every $x \in \mathbb{R}$, we have the following estimation.

$$
\begin{aligned}
&(B(w_m^\top x)v(w_m) - \varphi(x))^2 \\
&\leq 2B(w_m^\top x)^2 v(w_m)^2 + 2\varphi(x)^2 \\
&\leq 2B(w_m^\top x)^2 (L_v\|w_m - \mathbf{0}\|_2 + |v(\mathbf{0})|)^2 + 2\left(\mathbb{E}_{w\sim\mathcal{N}(0,I_d)}\left[B(w^\top x)v(w)\right]\right)^2 \\
&\leq 2B(w_m^\top x)^2 (2L_v^2\|w_m - \mathbf{0}\|_2^2 + 2|v(\mathbf{0})|^2) + 2\mathbb{E}_{w\sim\mathcal{N}(0,I_d)}\left[B(w^\top x)^2\right]\mathbb{E}_{w\sim\mathcal{N}(0,I_d)}\left[v(w)^2\right] \\
&\leq 4L_v^2\|w_m\|_2^2 + 4|v(\mathbf{0})|^2 + 2R^2,
\end{aligned}
$$

where we used the fact that $v$ is $L_v$-Lipschitz and $0 \leq B(w^\top x) \leq 1$.

Therefore, we have

$$
\begin{aligned}
&\mathbb{E}_W \exp\left(\lambda^2 (B(w_m^\top x)v(w_m) - \varphi(x))^2\right) \\
&\leq \mathbb{E}_W \exp\left(\lambda^2 (4L_v^2\|w_m\|_2^2 + 4|v(\mathbf{0})|^2 + 2R^2)\right) \\
&= \exp\left(\lambda^2 (4|v(\mathbf{0})|^2 + 2R^2)\right) \cdot \mathbb{E}_W \exp\left(4L_v^2\lambda^2\|w_m\|_2^2\right) \\
&= \exp\left(\lambda^2 (4|v(\mathbf{0})|^2 + 2R^2)\right) \cdot \prod_{i=1}^d \mathbb{E}_{w_{m,d}\sim\mathcal{N}(0,1)} \exp\left(4L_v^2\lambda^2 w_{m,d}^2\right) \\
&= \exp\left(\lambda^2 (4|v(\mathbf{0})|^2 + 2R^2)\right) \cdot \prod_{i=1}^d \frac{1}{\sqrt{1 - 8L_v^2\lambda^2}}
\end{aligned}
\tag{B.10}
$$

By applying $\frac{1}{1-x} \leq e^{2x}$ over $x \in [0, 1/2]$, we have that for $\lambda^2 \leq \frac{1}{16L_v^2}$,

$$
\begin{aligned}
&\exp\left(\lambda^2 (4|v(\mathbf{0})|^2 + 2R^2)\right) \cdot \prod_{i=1}^d \frac{1}{\sqrt{1 - 8L_v^2\lambda^2}} \\
&\leq \exp\left(\lambda^2 (4|v(\mathbf{0})|^2 + 2R^2)\right) \exp\left(8dL_v^2\lambda^2\right) \\
&= \exp\left(\lambda^2 (8dL_v^2 + 4|v(\mathbf{0})|^2 + 2R^2)\right) \\
&\leq \exp\left(\lambda^2 (16dL_v^2 + 4|v(\mathbf{0})|^2 + 2R^2)\right) \\
&\leq \exp\left(\lambda^2 \cdot 10R^2\right).
\end{aligned}
\tag{B.11}
$$

To summarize, let $Y_m = B(w_m^\top x)v(w_m) - \varphi(x)$, then for $\lambda^2 \leq 1/(10R^2)$, it holds that

$$
\mathbb{E}_W \exp\left(\lambda^2 Y_m^2\right) \leq \exp\left(\lambda^2 \cdot 10R^2\right).
$$

By Lemma A.4, we have that for all $\lambda \in \mathbb{R}$,

$$
\mathbb{E}_W \exp\left(\lambda Y_m\right) \leq \exp\left(\lambda^2 \cdot 10R^2\right).
\tag{B.12}
$$

Note that $Y_1, Y_2, ..., Y_M$ are independent. Therefore, we have

$$
\begin{aligned}
&\mathbb{E}_W \exp\left(\lambda\left(\frac{1}{M}\sum_{m=1}^M B(w_m^\top x)v(w_m) - \varphi(x)\right)\right) \\
&= \mathbb{E}_W \exp\left(\frac{\lambda}{M}\sum_{m=1}^M Y_m\right) = \prod_{m=1}^M \mathbb{E}_{w_m} \exp\left(\frac{\lambda}{M}Y_m\right) \\
&\leq \exp\left(\lambda^2 \cdot 10R^2/M\right).
\end{aligned}
\tag{B.13}
$$

By Lemma A.4 again, we have that for $\lambda^2 \leq M/(160R^2)$,

$$
\mathbb{E}_W \exp\left(\lambda^2\left(\frac{1}{M}\sum_{m=1}^M B(w_m^\top x)v(w_m) - \varphi(x)\right)^2\right)
\tag{B.14}
$$

$$
\leq \exp\left(160R^2\lambda^2/M\right).
$$

Taking expectation over $x$ on both sides and plugging it back to (B.9), we have that

$$\mathbb{E}_W \exp\left(\lambda^2 \left(\mathbb{E}_x \left| \frac{1}{M} \sum_{m=1}^M B(w_m^\top x) v(w_m) - \varphi(x) \right| \right)^2 \right) \leq \exp\left(160 R^2 \lambda^2 / M\right).$$

Because $\sqrt{2}\sqrt{160 R^2} \leq 18R$, by Lemma A.3, we conclude that

$$\left\| \mathbb{E}_x \left| \frac{1}{M} \sum_{m=1}^M B(w_m^\top x) v(w_m) - \varphi(x) \right| \right\|_{\psi_2} \leq \frac{18R}{\sqrt{M}}.$$

Consequently, applying Lemma A.2, for $\delta > 0$, by taking some $\epsilon = \frac{18R\sqrt{\log(4/\delta)}}{\sqrt{M}}$, we have that

$$P\left(\mathbb{E}_x |\hat\varphi(x) - \varphi(x)| \geq \epsilon\right)$$

$$= P\left(\mathbb{E}_x \left| \frac{1}{M} \sum_{m=1}^M B(w_m^\top x) v(w_m) - \varphi(x) \right| \geq \epsilon\right)$$

$$\leq 2\exp\left(-\frac{M\epsilon^2}{(18R)^2}\right) \leq \delta/2.$$

Hence, with probability of at least $1 - \delta/2$, it holds that

$$\mathbb{E}_x |\hat\varphi(x) - \varphi(x)| \leq \frac{18R\sqrt{\log(4/\delta)}}{\sqrt{M}}.$$

In the remaining part of the proof, we consider the high probability bound of $\sum_{m=1}^M v_m^2$. To start with, we show that $v(w)$ is a sub-gaussian random variable in which $w \sim \mathcal{N}(0, I_d)$.

$$\mathbb{E}\exp\left(\lambda^2 v(w)^2\right) \leq \mathbb{E}\exp\left(\lambda^2 (2L_v^2 \|w_m\|_2^2 + 2|v(\mathbf{0})|^2)\right)$$
$$\leq \exp\left((4L_v^2 d + 2|v(\mathbf{0})|^2)\lambda^2\right),$$

for $\lambda$ such that $(4L_v^2 d + 2|v(\mathbf{0})|^2)\lambda^2 \leq 1$. By Lemma A.3, we have $\|v(w)\|_{\psi_2}^2 \leq (4L_v^2 d + 2|v(\mathbf{0})|^2)/\log 2 \leq 4R^2$. Hence, by Lemma A.1, we have $\|v(w)^2\|_{\psi_1} = \|v(w)\|_{\psi_2}^2 \leq 4R^2$. By triangle inequality, we have $\|v(w)^2 - \mathbb{E}[v(w)^2]\|_{\psi_1} \leq \|v(w)^2\|_{\psi_1} + \|\mathbb{E}[v(w)^2]\|_{\psi_1}$. Given that $\mathbb{E}[v(w)^2]$ is a constant with an upper bound $R^2$, by the definition of the sub-exponential norm, we have $\|\mathbb{E}[v(w)^2]\|_{\psi_1} \leq \mathbb{E}[v(w)^2]/\log 2 \leq 2R^2$. To conclude, we have that $\|v(w)^2 - \mathbb{E}[v(w)^2]\|_{\psi_1} \leq 6R^2$.

We apply Lemma A.6 for random variables $X_m = v(w_m)^2 - \mathbb{E}[v(w)^2]$ by setting $t = 24R^2\left(\sqrt{\frac{\log(2/\delta)}{M}} + \frac{\log(2/\delta)}{M}\right)$. We obtain

$$P\left(\frac{1}{M} \sum_{m=1}^M v(w_m)^2 - \mathbb{E}[v(w)^2] > t\right) \leq \exp\left(-\min\left\{\frac{Mt^2}{16\|X\|_{\psi_1}^2}, \frac{Mt}{4\|X\|_{\psi_1}}\right\}\right) \leq \frac{\delta}{2}.$$

Because $\mathbb{E}[v(w)^2] \leq R^2$, we obtain that

$$P\left(\frac{1}{M} \sum_{m=1}^M v(w_m)^2 - R^2 > t\right) \leq P\left(\frac{1}{M} \sum_{m=1}^M v(w_m)^2 - \mathbb{E}[v(w)^2] > t\right) \leq \frac{\delta}{2}.$$

Therefore, with probability of at least $1 - \delta/2$, we have

$$\frac{1}{M} \sum_{m=1}^M v(w_m)^2 \leq R^2 + 24R^2\left(\sqrt{\frac{\log(2/\delta)}{M}} + \frac{\log(2/\delta)}{M}\right) \leq R^2 + 24R^2\left(\sqrt{\log(2/\delta)} + \log(2/\delta)\right).$$

Without loss of generality, we assume $\delta < 1/2$, then $1 < \sqrt{\log(2/\delta)} < \log(2/\delta)$ and hence

$$\frac{1}{M} \sum_{m=1}^M v(w_m)^2 \leq 49R^2 \log(2/\delta).$$

Combining the two inequalities and taking the union bound of the probabilities, we have that with probability of at least at least $1 - \delta$, it holds that

$$\mathbb{E}_x \left| \hat{\varphi}(x) - \varphi(x) \right| \leq \frac{18R\sqrt{\log(4/\delta)}}{\sqrt{M}},$$

and

$$\frac{1}{M} \sum_{m=1}^{M} v(w_m)^2 \leq 49R^2 \log(2/\delta).$$

$\square$

## C  DEFFERRED PROOF IN SECTION 4

### C.1  PROOF OF PROPOSITION 4.1

**Proof.** To start with, we define the Gaussian function with parameter $h$ as

$$\phi_h(x) := \frac{1}{\sqrt{2\pi}h} \exp\left(-\frac{x^2}{2h^2}\right).$$

First, we approximate $\sigma$ by $\sigma * \phi_h = \int_{\mathbb{R}} \sigma(x - y)\phi_h(y)dy$. Because $\sigma$ is $L$-Lipschitz continuous, we have that $|\sigma(x) - \sigma(x - y)| \leq L|y|$. Together with the fact $|\sigma| \leq \|\sigma\|_\infty$, we have that

$$|\sigma(x) - (\sigma * \phi_h)(x)|$$

$$= \left| \sigma(x) - \int_{\mathbb{R}} \sigma(x - y)\phi_h(y)dy \right|$$

$$\leq \int_{\mathbb{R}} |\sigma(x) - \sigma(x - y)|\, \phi_h(y)dy$$

$$= \int_{[-\delta,\delta]} |\sigma(x) - \sigma(x - y)|\, \phi_h(y)dy + \int_{\mathbb{R}-[-\delta,\delta]} |\sigma(x) - \sigma(x - y)|\, \phi_h(y)dy$$

$$\leq \int_{[-\delta,\delta]} L\,|y|\, \phi_h(y)dy + \int_{\mathbb{R}-[-\delta,\delta]} 2\|\sigma\|_\infty \phi_h(y)dy$$

$$\leq L\delta + 2\|\sigma\|_\infty \cdot P\left(|Z| \geq \frac{\delta}{h}\right),$$

where $Z \sim \mathcal{N}(0, 1)$. The tail probability of Gaussian random variable is estimated as

$$P\left(|Z| \geq \frac{\delta}{h}\right) = 2P\left(Z \geq \frac{\delta}{h}\right)$$

$$\overset{\lambda \geq 0}{=} 2P\left(e^{\lambda Z} \geq e^{\frac{\lambda\delta}{h}}\right)$$

$$\leq 2 \inf_{\lambda > 0} \frac{\mathbb{E}e^{\lambda Z}}{e^{\frac{\lambda\delta}{h}}}$$

$$= 2\exp\left(-\frac{\delta^2}{2h^2}\right).$$

By taking

$$\delta = \frac{\epsilon}{4L}, \quad h \leq \frac{\epsilon}{4\sqrt{2}L\sqrt{\log \frac{16\|\sigma\|_\infty}{\epsilon}}},$$

we have

$$\left| \sigma(x) - \int_{\mathbb{R}} \sigma(x - y)\phi_h(y)dy \right|$$

$$\leq L\delta + 2\|\sigma\|_\infty \cdot P\left(|Z| \geq \frac{\delta}{h}\right)$$

$$\leq \frac{\epsilon}{4} + \frac{\epsilon}{4} = \frac{\epsilon}{2}.$$

In the second step, we approximate $\sigma * \phi_h$ by the Riemann sum $\sum_{i=1}^{N} f(y_i) \cdot (y_i - y_{i-1}) \cdot \phi_h(x - y_i)$.

For the convolution part, we have

$$
\begin{aligned}
(\sigma * \phi_h)(x) &= \int_{\mathbb{R}} \sigma(x - y) \phi_h(y) dy \\
&= \int_{\mathbb{R}} \sigma(y) \phi_h(x - y) dy \\
&= \int_{\mathcal{K}} \sigma(y) \phi_h(x - y) dy \\
&= \sum_{i=1}^{N} \int_{y_{i-1}}^{y_i} \sigma(y) \phi_h(x - y) dy.
\end{aligned}
$$

Then we have

$$
\begin{aligned}
&\left| (\sigma * \phi_h)(x) - \sum_{i=1}^{N} \sigma(y_i) \cdot (y_i - y_{i-1}) \cdot \phi_h(x - y_i) \right| \\
&= \left| \sum_{i=1}^{N} \int_{y_{i-1}}^{y_i} \sigma(y) \phi_h(x - y) dy - \sum_{i=1}^{N} \int_{y_{i-1}}^{y_i} \sigma(y_i) \phi_h(x - y_i) dy \right| \\
&\leq \left| \sum_{i=1}^{N} \int_{y_{i-1}}^{y_i} \sigma(y) \phi_h(x - y) dy - \sum_{i=1}^{N} \int_{y_{i-1}}^{y_i} \sigma(y_i) \phi_h(x - y) dy \right| \\
&\quad + \left| \sum_{i=1}^{N} \int_{y_{i-1}}^{y_i} \sigma(y_i) \phi_h(x - y) dy - \sum_{i=1}^{N} \int_{y_{i-1}}^{y_i} \sigma(y_i) \phi_h(x - y_i) dy \right| \\
&\leq \sum_{i=1}^{N} \int_{y_{i-1}}^{y_i} |\sigma(y) - \sigma(y_i)| \phi_h(x - y) dy + \sum_{i=1}^{N} \int_{y_{i-1}}^{y_i} |\sigma(y_i)| \cdot |\phi_h(x - y) - \phi_h(x - y_i)| \, dy \\
&\leq \sum_{i=1}^{N} L(y_i - y_{i-1}) \int_{y_{i-1}}^{y_i} \phi_h(x - y) dy + \|\sigma\|_\infty \sum_{i=1}^{N} \int_{y_{i-1}}^{y_i} |\phi_h(x - y) - \phi_h(x - y_i)| \, dy.
\end{aligned}
\tag{C.1}
$$

For the first term, if $|\mathcal{K}|/N \leq \epsilon/4L$, then we have

$$
\sum_{i=1}^{N} L(y_i - y_{i-1}) \int_{y_{i-1}}^{y_i} \phi_h(x - y) dy \leq \frac{L|\mathcal{K}|}{N} \sum_{i=1}^{N} \int_{y_{i-1}}^{y_i} \phi_h(x - y) dy \leq \frac{L|\mathcal{K}|}{N} \leq \frac{\epsilon}{4}.
\tag{C.2}
$$

For the second term, we first consider the derivative of $\phi_h(x)$.

$$
\begin{aligned}
|\phi_h'(x)| &= \left| \frac{1}{\sqrt{2\pi}h^2} \cdot \frac{x}{h} \cdot \exp\left( -\frac{1}{2}\left(\frac{x}{h}\right)^2 \right) \right| \\
&\leq \frac{1}{\sqrt{2\pi}h^2} \exp\left( -\frac{1}{4}\left(\frac{x}{h}\right)^2 \right)
\end{aligned}
$$

where we use the inequality $x \leq \exp(x^2/4)$.

Taking $t = \sqrt{4 \log\left( \frac{8\|\sigma\|_\infty |\mathcal{K}|^2}{\sqrt{2\pi}} \cdot \frac{1}{\epsilon N h^2} \right)}$, if $|x| > th$, then

$$
|\phi_h'(x)| \leq \frac{\epsilon N}{8\|\sigma\|_\infty |\mathcal{K}|^2}.
$$

If $|x| \le th$, then

$$\begin{aligned}
|\phi_h'(x)| &= \left| \frac{1}{\sqrt{2\pi h^2}} \cdot \frac{x}{h} \cdot \exp\left(-\frac{1}{2}\left(\frac{x}{h}\right)^2\right) \right| \\
&\le \left| \frac{1}{\sqrt{2\pi h^2}} \sup_{t\in\mathbb{R}} \left\{ t\exp\left(-\frac{t^2}{2}\right) \right\} \right| \\
&= \frac{1}{\sqrt{2\pi e h^2}}.
\end{aligned}$$

Consequently, for the second term, it holds that

$$\begin{aligned}
&\|\sigma\|_\infty \sum_{i=1}^N \int_{y_{i-1}}^{y_i} |\phi_h(x-y) - \phi_h(x-y_i)|\, dy \\
={}& \|\sigma\|_\infty \sum_{i=1}^N \int_{y_{i-1}}^{y_i} \left| \int_y^{y_i} \phi_h'(x-z)\, dz \right| dy \\
\le{}& \|\sigma\|_\infty \sum_{i=1}^N \int_{y_{i-1}}^{y_i} \left| \sup_{z\in[x-y_i,\,x-y_{i-1}]} |\phi_h'(z)||y-y_i| \right| dy \\
\le{}& \|\sigma\|_\infty \sum_{i=1}^N \sup_{z\in[x-y_i,\,x-y_{i-1}]} |\phi_h'(z)| \left(\frac{|\mathcal{K}|}{N}\right)^2 & \text{(C.3)} \\
\le{}& \|\sigma\|_\infty \frac{2th}{\frac{|\mathcal{K}|}{N}} \cdot \frac{1}{\sqrt{2\pi e h^2}} \cdot \left(\frac{|\mathcal{K}|}{N}\right)^2 + \|\sigma\|_\infty N \cdot \frac{\epsilon N}{8\|\sigma\|_\infty |\mathcal{K}|^2} \cdot \left(\frac{|\mathcal{K}|}{N}\right)^2 \\
={}& \|\sigma\|_\infty \sqrt{\frac{2}{\pi e}} \cdot \frac{t|\mathcal{K}|}{hN} + \frac{\epsilon}{8} \\
={}& \|\sigma\|_\infty \sqrt{\frac{2}{\pi e}} \sqrt{4\log\left(\frac{8\|\sigma\|_\infty |\mathcal{K}|}{\sqrt{2\pi}} \cdot \frac{|\mathcal{K}|}{\epsilon h^2 N}\right)} \cdot \frac{|\mathcal{K}|}{hN} + \frac{\epsilon}{8}.
\end{aligned}$$

The fifth line holds because there are at most $2thN/|\mathcal{K}|$ intervals in which $|\phi_h'| > \frac{\epsilon N}{8\|\sigma\|_\infty |\mathcal{K}|^2}$.

Let

$$\frac{|\mathcal{K}|}{N} \le \frac{\epsilon h\sqrt{\pi e}}{16\sqrt{2}\|\sigma\|_\infty \log\left(\frac{8\|\sigma\|_\infty |\mathcal{K}|}{\sqrt{2\pi}\epsilon h^2}\right)} \wedge \frac{\epsilon}{4L} \ll 1.$$

Then

$$\begin{aligned}
&\|\sigma\|_\infty \sqrt{\frac{2}{\pi e}} \sqrt{4\log\left(\frac{8\|\sigma\|_\infty |\mathcal{K}|}{\sqrt{2\pi}} \cdot \frac{|\mathcal{K}|}{\epsilon h^2 N}\right)} \cdot \frac{|\mathcal{K}|}{hN} \\
={}& \|\sigma\|_\infty \sqrt{\frac{2}{\pi e}} \sqrt{4\log\left(\frac{8\|\sigma\|_\infty |\mathcal{K}|}{\sqrt{2\pi}\epsilon h^2}\right) + 4\log\left(\frac{|\mathcal{K}|}{N}\right)} \cdot \frac{|\mathcal{K}|}{hN} & \text{(C.4)} \\
\le{}& \|\sigma\|_\infty \sqrt{\frac{2}{\pi e}} \sqrt{4\log\left(\frac{8\|\sigma\|_\infty |\mathcal{K}|}{\sqrt{2\pi}\epsilon h^2}\right)} \cdot \frac{|\mathcal{K}|}{hN} \le \frac{\epsilon}{8}.
\end{aligned}$$

Putting (C.2), (C.3) and (C.4) into (C.1), we conclude that

$$\left| (\sigma * \phi_h)(x) - \sum_{i=1}^N \sigma(y_i) \cdot (y_i - y_{i-1}) \cdot \phi_h(x - y_i) \right| \le \frac{\epsilon}{2}.$$

Hence,

$$\left| \sigma(x) - \sum_{i=1}^{N} \sigma(y_i) \cdot (y_i - y_{i-1}) \cdot \phi_h(x - y_i) \right|$$

$$\leq |\sigma(x) - (\sigma * \phi_h)(x)| + \left| (\sigma * \phi_h)(x) - \sum_{i=1}^{N} \sigma(y_i) \cdot (y_i - y_{i-1}) \cdot \phi_h(x - y_i) \right|$$

$$\leq \frac{\epsilon}{2} + \frac{\epsilon}{2} = \epsilon.$$

Let

$$B_i(x) = \exp\left( -\frac{(x - y_i)^2}{2h^2} \right), \quad a_i = \frac{|\mathcal{K}|}{\sqrt{2\pi}hN} \cdot \sigma(y_i),$$

then

$$\sum_{i=1}^{N} \sigma(y_i) \cdot (y_i - y_{i-1}) \cdot \phi_h(x - y_i) = \sum_{i=1}^{N} a_i B_i(x).$$

Hence

$$\left\| \sigma - \sum_{i=1}^{N} a_i B_i(x) \right\|_\infty \leq \epsilon,$$

and

$$\sum_{i=1}^{N} |a_i| \leq \sum_{i=1}^{N} \frac{|\mathcal{K}|}{\sqrt{2\pi}hN} \cdot |\sigma(y_i)| \leq \frac{\|\sigma\|_\infty |\mathcal{K}|}{\sqrt{2\pi}h}.$$

In addition,

$$\sum_{i=1}^{N} |a_i|^2 \leq \sum_{i=1}^{N} \frac{|\mathcal{K}|^2}{2\pi h^2 N^2} \cdot |\sigma(y_i)|^2 \leq \frac{\|\sigma\|_\infty^2 |\mathcal{K}|}{2\pi} \cdot \frac{|\mathcal{K}|}{Nh^2}.$$

To conclude, if one sets

$$h_i \equiv h \leq \frac{\epsilon}{4\sqrt{2}L\sqrt{\log \frac{16\|\sigma\|_\infty}{\epsilon}}}, \quad \frac{|\mathcal{K}|}{N} \leq \frac{\epsilon h\sqrt{\pi e}}{16\sqrt{2}\|\sigma\|_\infty \log\left( \frac{8\|\sigma\|_\infty |\mathcal{K}|}{\sqrt{2\pi}\epsilon h^2} \right)} \wedge \frac{\epsilon}{4L}, \qquad (\text{C.5})$$

and $c_i$ be the grid points of $\mathcal{K}$, then there exists $\{a_i\}_{i=1}^{N}$ such that

$$\left\| \sigma - \sum_{i=1}^{N} a_i B_i(x) \right\|_\infty \leq \epsilon,$$

and

$$\sum_{i=1}^{N} |a_i| \leq \frac{\|\sigma\|_\infty |\mathcal{K}|}{\sqrt{2\pi}h}, \qquad \sum_{i=1}^{N} |a_i|^2 \leq \frac{\|\sigma\|_\infty^2 |\mathcal{K}|}{2\pi} \cdot \frac{|\mathcal{K}|}{Nh^2}.$$

We remark that the choice of $c_i$ could be arbitrary as long as $c_i \in [y_{i-1}, y_i]$.

Now, replacing $\epsilon$ with $\epsilon/R$ in (C.5), there exists $N > 0$ and $\{h_i, c_i, a_i\}_{i=1}^{N}$ such that

$$\left\| \sigma(x) - \sum_{i=1}^{N} a_i B_i(x) \right\|_\infty < \frac{\epsilon}{R}.$$

Thus

$$
\begin{aligned}
\left\| f^*(x) - \tilde{f}(x) \right\|_\infty &= \left\| \mathbb{E}_{w \sim \mathcal{N}(0,1)} \left[ \left( \sigma(w^\top x) - \sum_{i=1}^N a_i B_i(w^\top x) \right) v(w) \right] \right\|_\infty \\
&\leq \mathbb{E}_{w \sim \mathcal{N}(0,1)} \left[ \left\| \sigma(w^\top x) - \sum_{i=1}^N a_i B_i(w^\top x) \right\|_\infty |v(w)| \right] \\
&\leq \left\| \sigma(w^\top x) - \sum_{i=1}^N a_i B_i(w^\top x) \right\|_\infty \left( \mathbb{E}_{w \sim \mathcal{N}(0,1)} \left[ v(w)^2 \right] \right)^{\frac{1}{2}} \\
&\leq \frac{\epsilon}{R} \cdot R \leq \epsilon.
\end{aligned}
\tag{C.6}
$$

$\square$

### C.2 PROOF OF THEOREM 4.2

**Proof.**

For all $\epsilon > 0$, under the parameter settings of Proposition 4.1, there exists $\{a_i\}_{i=1}^N$ such that

$$
\left\| \tilde{f}(x) - f^*(x) \right\|_\infty \leq \epsilon, \quad \left\| \sigma(x) - \sum_{i=1}^N a_i B_i(x) \right\|_\infty < \epsilon/R, \quad \sum_{i=1}^N a_i^2 \leq \frac{\|\sigma\|_\infty^2 |\mathcal{K}|^2}{2\pi h^2 N}.
$$

So we first have

$$
\begin{aligned}
\mathbb{E}_x \left| \hat{f}(x) - f^*(x) \right| &= \mathbb{E}_x \left| \hat{f}(x) - \tilde{f}(x) + \tilde{f}(x) - f^*(x) \right| \\
&\leq \mathbb{E}_x \left| \hat{f}(x) - \tilde{f}(x) \right| + \mathbb{E}_x \left| \tilde{f}(x) - f^*(x) \right| \\
&\leq \mathbb{E}_x \left| \sum_{i=1}^N a_i \left( \hat{\varphi}_i(x) - \varphi_i(x) \right) \right| + \epsilon.
\end{aligned}
\tag{C.7}
$$

Next, we aim to derive a high probability bound on $\mathbb{E}_x \left| \sum_{i=1}^N a_i \left( \hat{\varphi}_i(x) - \varphi_i(x) \right) \right|$. The proof techniques are similar to those of Theorem 3.4. First, we have

$$
\begin{aligned}
\sum_{i=1}^N a_i \left( \hat{\varphi}_i(x) - \varphi_i(x) \right) &= \sum_{i=1}^N a_i \left( \frac{1}{M} \sum_{m=1}^M B_i(w_m^\top x) v(w_m) - \varphi_i(x) \right) \\
&= \frac{1}{M} \sum_{m=1}^M \left( \sum_{i=1}^N a_i B_i(w_m^\top x) v(w_m) - \sum_{i=1}^N a_i \varphi_i(x) \right)
\end{aligned}
$$

It boils down to estimating the sub-gaussian norms of the random variables $Z_m = \sum_{i=1}^N a_i B_i(w_m^\top x) v(w_m) - \sum_{i=1}^N a_i \varphi_i(x)$ where $\{w_m\}_{m \in [M]} \overset{i.i.d.}{\sim} \mathcal{N}(0, I_d)$.

Consider

$$
\begin{aligned}
Z_m^2 &= \left( \sum_{i=1}^N a_i B_i(w_m^\top x) v(w_m) - \sum_{i=1}^N a_i \varphi_i(x) \right)^2 \\
&\leq 2 \left( \sum_{i=1}^N a_i B_i(w_m^\top x) v(w_m) \right)^2 + 2 \left( \sum_{i=1}^N a_i \varphi_i(x) \right)^2 \\
&= 2v(w_m)^2 \left( \sum_{i=1}^N a_i B_i(w_m^\top x) \right)^2 + 2 \left( \mathbb{E}_w \sum_{i=1}^N a_i B_i(w^\top x) v(w) \right)^2 \\
&\leq 2v(w_m)^2 \left( \sum_{i=1}^N a_i B_i(w_m^\top x) \right)^2 + 2\mathbb{E}_w \left( \sum_{i=1}^N a_i B_i(w^\top x) \right)^2 \mathbb{E}_w \left( v(w)^2 \right).
\end{aligned}
$$

Because $\left\|\sigma(x) - \sum_{i=1}^{N} a_i B_i(x)\right\|_{\infty} < \epsilon/R$, we have $\left|\sum_{i=1}^{N} a_i B_i(x)\right| \leq \|\sigma\|_{\infty} + \epsilon/R$ for all $x$. Hence,

$$
\begin{aligned}
Z_m^2 &\leq 2v(w_m)^2 \left(\|\sigma\|_{\infty} + \epsilon/R\right)^2 + 2R^2 \left(\|\sigma\|_{\infty} + \epsilon/R\right)^2 \\
&\leq 2(L_v\|w_m\| + |v(\mathbf{0})|)^2 \left(\|\sigma\|_{\infty} + \epsilon/R\right)^2 + 2R^2 \left(\|\sigma\|_{\infty} + \epsilon/R\right)^2 \\
&\leq \left(4L_v^2\|w_m\|^2 + 4|v(\mathbf{0})|^2 + 2R^2\right)\left(\|\sigma\|_{\infty} + \epsilon/R\right)^2.
\end{aligned}
$$

Similar to the estimation in Eq. (B.10) and Eq. (B.11), we have that for $\lambda$ such that $10(\|\sigma\|_{\infty}R + \epsilon)^2\lambda^2 \leq 1$, it holds that

$$
\mathbb{E}_W e^{\lambda^2 Z_m^2} \leq e^{10(\|\sigma\|_{\infty}R+\epsilon)^2\lambda^2}.
$$

Similar to the estimatio in Eq. (B.12), Eq. (B.13) and Eq. (B.14), we have that for $\lambda$ such that $160(\|\sigma\|_{\infty}R + \epsilon)^2\lambda^2 \leq M$, it holds that

$$
\mathbb{E}_W e^{\lambda^2 \left(\sum_{m=1}^{M} Z_m/M\right)^2} \leq e^{160(\|\sigma\|_{\infty}R+\epsilon)^2\lambda^2/M}.
$$

Hence, similar to Eq. (B.9), we have

$$
\begin{aligned}
\mathbb{E}_W e^{\lambda^2 \left(\mathbb{E}_x \left|\sum_{i=1}^{N} a_i(\hat{\varphi}_i(x) - \varphi_i(x))\right|\right)^2} &\leq \mathbb{E}_x \mathbb{E}_W e^{\lambda^2 \left(\left|\sum_{i=1}^{N} a_i(\hat{\varphi}_i(x) - \varphi_i(x))\right|\right)^2} \\
&= \mathbb{E}_x \mathbb{E}_W e^{\lambda^2 \left(\sum_{m=1}^{M} Z_m/M\right)^2} \\
&\leq e^{160(\|\sigma\|_{\infty}R+\epsilon)^2\lambda^2/M}.
\end{aligned}
$$

By Lemma A.3, we obtain $\left\|\mathbb{E}_x \left|\sum_{i=1}^{N} a_i(\hat{\varphi}_i(x) - \varphi_i(x))\right|\right\|_{\psi_2} \leq 18(\|\sigma\|_{\infty}R + \epsilon)/\sqrt{M}$. By Lemma A.2, we have that

$$
P\left(\mathbb{E}_x \left|\sum_{i=1}^{N} a_i(\hat{\varphi}_i(x) - \varphi_i(x))\right| \geq \frac{18(\|\sigma\|_{\infty}R + \epsilon)\sqrt{\log(4/\delta)}}{\sqrt{M}}\right) \leq \frac{\delta}{2}.
$$

Namely, with probability of at least $1 - \delta/2$, we have

$$
\mathbb{E}_x \left|\sum_{i=1}^{N} a_i(\hat{\varphi}_i(x) - \varphi_i(x))\right| \leq \frac{18(\|\sigma\|_{\infty}R + \epsilon)\sqrt{\log(4/\delta)}}{\sqrt{M}}.
$$

Therefore, putting it back to (C.7), with probability of at least $1 - \delta/2$, we have

$$
\mathbb{E}_x \left|\hat{f}(x) - f^*(x)\right| \leq \frac{18(\|\sigma\|_{\infty}R + \epsilon)\sqrt{\log(4/\delta)}}{\sqrt{M}} + \epsilon.
$$

Further, by the proof of Theorem 3.4, the event

$$
\frac{1}{M}\sum_{m=1}^{M} v_m^2 \leq 49R^2\log(2/\delta)
$$

happens with probability of at least $1 - \delta/2$.

Taking the union bounds of the probability, we conclude that with probability of at least $1 - \delta$, the inequalities hold:

$$
\mathbb{E}_x \left|\hat{f}(x) - f^*(x)\right| \leq \frac{18(\|\sigma\|_{\infty}R + \epsilon)\sqrt{\log(4/\delta)}}{\sqrt{M}} + \epsilon,
$$

and

$$
\frac{1}{M}\sum_{m=1}^{M} v_m^2 \leq 49R^2\log(2/\delta).
$$

$\square$

# D    DEFERRED PROOF IN SECTION 5

We use Rademacher complexity to obtain the result in Theorem 5.1. We first recall the definition of Rademacher complexity. Suppose we are given samples $S = \{z_i = (x_i, y_i)\}_{i=1}^n$. Let

$$\ell \circ f_{\mathcal{V}} := \{(x, y) \mapsto \ell(f(x), y) : f \in f_{\mathcal{V}}\}$$

be the function class. Let

$$f_{\mathcal{V}} \circ S := \{(f(x_1), ..., f(x_n)) : f \in f_{\mathcal{V}}\},$$

$$\ell \circ f_{\mathcal{V}} \circ S := \{(\ell(f(x_1), y_1), ..., \ell(f(x_n), y_n)) : f \in f_{\mathcal{V}}\}$$

be vector sets. The Rademacher complexity of a function class $\mathcal{H}$ with respect to $S$ is defined as

$$\mathcal{R}(\mathcal{H} \circ S) := \frac{1}{n} \mathbb{E}_{\boldsymbol{\xi}} \sup_{h \in \mathcal{H}} \sum_{i=1}^n \xi_i h(z_i),$$

where $\boldsymbol{\xi} = (\xi_1, ..., \xi_n)$ and $\{\xi_i\}_{i \in [n]}$ are independent symmetric Bernoulli random variables.

Next, we introduce three lemmas for proving Theorem 5.1. The first one is a technical tool.

**Lemma D.1 (Talagrand's contraction principle (e.g., Exercise 6.7.7 in (Vershynin, 2018)))**
*Consider a bounded subset $T \subset \mathbb{R}^n$, and let $\{\xi_i\}_{i \in [n]}$ be independent symmetric Bernoulli random variables. If $\phi_i : \mathbb{R} \to \mathbb{R}$ are $\rho$-Lipschitz functions, then*

$$\mathbb{E}_{\boldsymbol{\xi}} \sup_{t \in T} \sum_{i=1}^n \xi_i \phi_i(t_i) \le \rho \, \mathbb{E}_{\boldsymbol{\xi}} \sup_{t \in T} \sum_{i=1}^n \xi_i t_i.$$

Then, through Lemma D.1, we can obtain the following result describing the Rademacher complexity of the function class of interests.

**Lemma D.2** *All $f \in f_{\mathcal{V}}$ are bounded:*

$$\|f\|_\infty \le \frac{7\|\sigma\|_\infty |\mathcal{K}| R \sqrt{\log(2/\delta)}}{h\sqrt{2\pi}}.$$

*Furthermore, the Rademacher complexity of $\ell \circ f_{\mathcal{V}}$ with respect to samples $S$ is bounded as*

$$\mathcal{R}(\ell \circ f_{\mathcal{V}} \circ S) \le \frac{7\rho\|\sigma\|_\infty |\mathcal{K}| R \sqrt{\log(2/\delta)/2\pi}}{h\sqrt{n}}.$$

For the coherence of the statements, we give the proof of Lemma D.2 at the end of this section. Finally, we derive the excess risk from the Rademacher complexity using the well known result in supervised learning illustrated below.

**Lemma D.3 (e.g., Theorem 26.5 in (Shalev-Shwartz & Ben-David, 2014))** *Assume that for all $z = (x, y) \sim \mathrm{P}$ and $f \in f_{\mathcal{V}}$ we have that $|\ell(f(x), y)| \le c$. Then for any $\hat{f} \in f_{\mathcal{V}}$, with probability of at least $1 - \delta$ over $\{(x_i, y_i)\}_{i \in [n]} \overset{i.i.d}{\sim} \mathrm{P}$, it holds that*

$$L_D(f_S) - L_D(\hat{f}) \le 2\mathcal{R}(\ell \circ f_{\mathcal{V}} \circ S) + 5c\sqrt{\frac{2\log(8/\delta)}{n}}.$$

**Formal proof of Theorem 5.1.**    Under the conditions and parameter settings of $h, N, \{c_i\}_{i=1}^N$ in Theorem 4.2, with probability of at least $1 - \delta$ over $W = (w_1, ..., w_M)$, there exists $\hat{f} \in f_{\mathcal{V}}$ such that

$$\mathbb{E}_x \left| \hat{f}(x) - f^*(x) \right| \le \frac{18(\|\sigma\|_\infty R + \epsilon)\sqrt{\log(4/\delta)}}{\sqrt{M}} + \epsilon.$$

On the other hand, for all $f \in f_{\mathcal{V}}$ and $(x, y)$, we have that

$$|\ell(f(x), y)| \le |\ell(0, y)| + \rho|f(x) - 0| \le \rho \left(1 + \frac{7\|\sigma\|_\infty |\mathcal{K}| R \sqrt{\log(2/\delta)}}{h\sqrt{2\pi}}\right) =: c,$$

where we use the first part of Lemma D.2, the Lipschitz property of $\ell$ and the relation $|\ell(0, y)| \le \rho$.

Apply the second part of Lemma D.2 and D.3 for $\hat{f}$. Then with probability of at least $1 - \delta$ over $\{(x_i, y_i)\}_{i \in [n]} \overset{i.i.d}{\sim} \mathrm{P}$, we have that

$$L_D(f_S) - L_D(\hat{f}) \le 2\mathcal{R}(\ell \circ f_\mathcal{V} \circ S) + 5c\sqrt{\frac{2\log(8/\delta)}{n}}$$

$$\le 2\rho\frac{7\|\sigma\|_\infty|\mathcal{K}|R\sqrt{\log(2/\delta)/2\pi}}{h}\sqrt{\frac{1}{n}} + 5\rho\left(1 + \frac{7\|\sigma\|_\infty|\mathcal{K}|R\sqrt{\log(2/\delta)/2\pi}}{h}\right)\sqrt{\frac{2\log(8/\delta)}{n}}$$

$$\le 7\rho\left(1 + \frac{7\|\sigma\|_\infty|\mathcal{K}|R\sqrt{\log(2/\delta)}}{h}\right)\sqrt{\frac{2\log(8/\delta)}{n}}.$$

Next, we notice that with probability of at least $1 - \delta$ over $W = (w_1, ..., w_M)$,

$$L_D(\hat{f}) - L_D(f^*) = \mathbb{E}_{x, y \sim \mathrm{P}}[\ell(\hat{f}(x), y) - \ell(f^*(x), y)]$$

$$\le \rho\,\mathbb{E}_x\left|\hat{f}(x) - f^*(x)\right|$$

$$\le \rho\frac{18(\|\sigma\|_\infty R + \epsilon)\sqrt{\log(4/\delta)}}{\sqrt{M}} + \rho\epsilon.$$

Combining the two inequalities and taking the union bounds of the probabilities, we conclude that with probability of at least $1 - 2\delta$ over $W$ and $S$, it holds that

$$L_D(f_S) - L_D(f^*) \le 7\rho\left(1 + \frac{7\|\sigma\|_\infty|\mathcal{K}|R\sqrt{\log(2/\delta)}}{h}\right)\sqrt{\frac{2\log(8/\delta)}{n}} + \rho\frac{18(\|\sigma\|_\infty R + \epsilon)\sqrt{\log(4/\delta)}}{\sqrt{M}} + \rho\epsilon.$$

Without loss of generality, assume $h \le 1$ and $\delta \le 1/2$, then $1 \le \sqrt{\log(2/\delta)}/h$, $\sqrt{\log(2/\delta)} \le \sqrt{2\log(8/\delta)}$. Consequently,

$$L_D(f_S) - L_D(f^*) \le \rho\frac{14\left(1 + 7\|\sigma\|_\infty|\mathcal{K}|R\right)\log(8/\delta)}{h\sqrt{n}} + \rho\frac{18(\|\sigma\|_\infty R + \epsilon)\sqrt{\log(4/\delta)}}{\sqrt{M}} + \rho\epsilon.$$

Replacing $\delta$ with $\delta/2$, with probability of at least $1 - \delta$, we have

$$L_D(f_S) - L_D(f^*) \le \rho\frac{14\left(1 + 7\|\sigma\|_\infty|\mathcal{K}|R\right)\log(16/\delta)}{h\sqrt{n}} + \rho\frac{18(\|\sigma\|_\infty R + \epsilon)\sqrt{\log(8/\delta)}}{\sqrt{M}} + \rho\epsilon.$$

Let $C = \max\{14\left(1 + 7\|\sigma\|_\infty|\mathcal{K}|R\right), 18(\|\sigma\|_\infty R + \epsilon)\}$, we obtain that

$$L_D(f_S) - L_D(f^*) \le \frac{\rho C\log(16/\delta)}{h\sqrt{n}} + \frac{\rho C\sqrt{\log(8/\delta)}}{\sqrt{M}} + \rho\epsilon.$$

$\square$

At the end of the proof, we supplement the proof of the second lemma. The proof of Lemma D.1 and D.3 can be found readily in the literature and are hence omitted.

**Proof of Lemma D.2.** Let $\phi_i(t) = \ell(t, y_i)$ and $t_i = f(x_i)$. Then $\phi_i(t)$ is $\rho$-Lipschitz continuous with respect to $t$. For the boundedness of $T = \{(f(x_1), ..., f(x_n)) : f \in f_\mathcal{V}\}$, we can see that for

all $f \in f_{\mathcal{V}}$, it holds that

$$
\begin{aligned}
|f| &= \left| \frac{1}{M} \sum_{k=1}^{N} a_k \sum_{m=1}^{M} B_k(w_m^\top x) v_m \right| \\
&\leq \frac{1}{M} \sqrt{\sum_{k=1}^{N} a_k^2} \cdot \sqrt{\sum_{k=1}^{N} \left( \sum_{m=1}^{M} B_k(w_m^\top x) v_m \right)^2} \\
&\leq \frac{1}{M} \sqrt{\sum_{k=1}^{N} a_k^2} \cdot \sqrt{\sum_{k=1}^{N} \left( \sum_{m=1}^{M} B_k^2(w_m^\top x) \sum_{m=1}^{M} v_m^2 \right)} \\
&\leq \frac{1}{M} \|\boldsymbol{a}\|_2 \cdot \sqrt{NM} \|\boldsymbol{v}\|_2 \\
&\leq \sqrt{\frac{N}{M}} \frac{\|\sigma\|_\infty |\mathcal{K}|}{h\sqrt{2\pi N}} \cdot \sqrt{49MR^2 \log\left(\frac{2}{\delta}\right)} \\
&= \frac{7\|\sigma\|_\infty |\mathcal{K}| R \sqrt{\log(2/\delta)}}{h\sqrt{2\pi}}.
\end{aligned}
\tag{D.1}
$$

Hence,

$$
\|f\|_\infty \leq \frac{7\|\sigma\|_\infty |\mathcal{K}| R \sqrt{\log(2/\delta)}}{h\sqrt{2\pi}},
$$

and for all $t \in T$, $t = (f(x_1), ..., f(x_n))$ and $\|t\| \leq \sqrt{n} \|f\|_\infty \leq \frac{7\|\sigma\|_\infty |\mathcal{K}| R \sqrt{n \log(2/\delta)}}{h\sqrt{2\pi}}$.

By applying Lemma D.1, we have

$$
\mathbb{E}_{\boldsymbol{\xi}} \sup_{t \in T} \sum_{i=1}^{n} \xi_i \ell(f(x_i), y_i) \leq \rho \, \mathbb{E}_{\boldsymbol{\xi}} \sup_{t \in T} \sum_{i=1}^{n} \xi_i f(x_i).
$$

To continue, let $K_1 = \frac{\|\sigma\|_\infty |\mathcal{K}|}{h\sqrt{2\pi N}}$, $K_2 = \sqrt{49MR^2 \log\left(\frac{2}{\delta}\right)}$, $\mathbf{B}_i \in \mathbb{R}^{N \times M}$ with $(\mathbf{B}_i)_{k,m} = B_k(w_m^\top x_i)$, then we have

$$
\begin{aligned}
&\mathbb{E}_{\boldsymbol{\xi}} \sup_{f \in f_{\mathcal{V}}} \sum_{i=1}^{n} \xi_i f(x_i) \\
=&\mathbb{E}_{\boldsymbol{\xi}} \sup_{\substack{\|\boldsymbol{a}\|_2 \leq K_1 \\ \|\boldsymbol{v}\|_2 \leq K_2}} \sum_{i=1}^{n} \xi_i \frac{1}{M} \sum_{k=1}^{N} a_k \sum_{m=1}^{M} B_k(w_m^\top x_i) v_m \\
=&\frac{1}{M} \mathbb{E}_{\boldsymbol{\xi}} \sup_{\substack{\|\boldsymbol{a}\|_2 \leq K_1 \\ \|\boldsymbol{v}\|_2 \leq K_2}} \sum_{k=1}^{N} a_k \sum_{m=1}^{M} \left( \sum_{i=1}^{n} \xi_i B_k(w_m^\top x_i) \right) v_m \\
=&\frac{1}{M} \mathbb{E}_{\boldsymbol{\xi}} \sup_{\substack{\|\boldsymbol{a}\|_2 \leq K_1 \\ \|\boldsymbol{v}\|_2 \leq K_2}} \boldsymbol{a}^\top \left( \sum_{i=1}^{n} \xi_i \mathbf{B}_i \right) \boldsymbol{v},
\end{aligned}
$$

Let $\|\cdot\|$ be the operator norm of a matrix, namely the largest singular value of a matrix. Then by the equivalent definition of the operator norm, we have that

$$
\frac{1}{M} \mathbb{E}_{\boldsymbol{\xi}} \sup_{\substack{\|\boldsymbol{a}\|_2 \leq K_1 \\ \|\boldsymbol{v}\|_2 \leq K_2}} \boldsymbol{a}^\top \left( \sum_{i=1}^{n} \xi_i \mathbf{B}_i \right) \boldsymbol{v} = \frac{K_1 K_2}{M} \mathbb{E}_{\boldsymbol{\xi}} \left\| \sum_{i=1}^{n} \xi_i \mathbf{B}_i \right\|.
$$

Furthermore, we have that for any matrix $\mathbf{A}$, it holds that

$$
\|\mathbf{A}\| \leq \|\mathbf{A}\|_{\mathrm{Fr}} = \sqrt{\mathrm{Tr}\left(\mathbf{A}\mathbf{A}^\top\right)}.
$$

Plugging it into the former expression, we have

$$
\begin{aligned}
\frac{K_1 K_2}{M} \mathbb{E}_{\boldsymbol{\xi}} \left\| \sum_{i=1}^n \xi_i \mathbf{B}_i \right\| &\leq \frac{K_1 K_2}{M} \mathbb{E}_{\boldsymbol{\xi}} \left\| \sum_{i=1}^n \xi_i \mathbf{B}_i \right\|_{\mathrm{Fr}} \\
&\leq \frac{K_1 K_2}{M} \sqrt{\mathbb{E}_{\boldsymbol{\xi}} \left\| \sum_{i=1}^n \xi_i \mathbf{B}_i \right\|_{\mathrm{Fr}}^2} \\
&= \frac{K_1 K_2}{M} \sqrt{\mathbb{E}_{\boldsymbol{\xi}} \mathrm{Tr} \left( \sum_{i=1}^n \xi_i \mathbf{B}_i \right) \left( \sum_{i=1}^n \xi_i \mathbf{B}_i \right)^\top} \\
&= \frac{K_1 K_2}{M} \sqrt{\mathrm{Tr} \mathbb{E}_{\boldsymbol{\xi}} \left( \sum_{i=1}^n \xi_i^2 \mathbf{B}_i \mathbf{B}_i^\top + \sum_{\substack{i \neq j \\ i,j \in [n]}} \xi_i \xi_j \mathbf{B}_i \mathbf{B}_j^\top \right)} \\
&= \frac{K_1 K_2}{M} \sqrt{\mathrm{Tr} \left( \sum_{i=1}^n \mathbf{B}_i \mathbf{B}_i^\top \right)} \\
&= \frac{K_1 K_2}{M} \sqrt{\sum_{i=1}^n \| \mathbf{B}_i \|_{\mathrm{Fr}}^2} \\
&\leq \frac{K_1 K_2}{M} \sqrt{nNM} = \frac{7 \|\sigma\|_\infty |\mathcal{K}| R \sqrt{n \log(2/\delta)}}{h \sqrt{2\pi}}.
\end{aligned}
$$

Finally, we conclude that

$$
\begin{aligned}
\mathcal{R}(\ell \circ f_{\mathcal{V}} \circ S) &= \frac{1}{n} \mathbb{E}_{\boldsymbol{\xi}} \sup_{t \in T} \sum_{i=1}^n \xi_i \ell(f(x_i), y_i) \\
&\leq \frac{\rho}{n} \mathbb{E}_{\boldsymbol{\xi}} \sup_{t \in T} \sum_{i=1}^n \xi_i f(x_i) \\
&\leq \frac{7 \rho \|\sigma\|_\infty |\mathcal{K}| R \sqrt{\log(2/\delta)/2\pi}}{h \sqrt{n}}.
\end{aligned}
$$

$\square$

## E  FURTHER DETAILS ON EXPERIMENTS

**Datasets.**  To create datasets, we sampled $10^5$ values of $w$ and using the empirical average $\sum_{m=1}^{10^5} \sigma_i(w_m^\top x) v_i(w_m)/10^5$ to approximate $f_i(x)$, so that the approximation error is around $C * 10^{-3}$. We sampled $\{x_i\}_{i \in [n]} \overset{i.i.d.}{\sim} \mathcal{N}(0, I_d)$ for sample size $n = 15000$ and $d = 2$.

**Optimization details.**  We form the learning problem (5.3) as a unconstrained optimization problem:

$$
\min_{\boldsymbol{a}, \boldsymbol{v}} \frac{1}{n} \sum_{i=1}^n (\boldsymbol{a}^\top \mathbf{B}(x_i) \boldsymbol{v} - y_i)^2 + \lambda_1 (\|\boldsymbol{a}\|_2^2 - \|\boldsymbol{v}\|_2^2)^2 + \lambda_2 \|\boldsymbol{a}\|_1, \tag{E.1}
$$

where $(\mathbf{B}(x_i))_{k,m} = B_k(w_m^\top x_i)$. The problem is categorized as matrix sensing, a canonical optimization problem in low-rank matrix factorization (Chi et al., 2019). The first regularizer is necessary to balance the size of the two vectors and to guarantee convergence. The second is the common $L_1$ regularizer that aims to obtain $\boldsymbol{a}$ with sparse components. We use Adam to train the model. The codes are available at `https://github.com/3b6bf22/repo`.

**Experiments on a wider range of activation functions.**

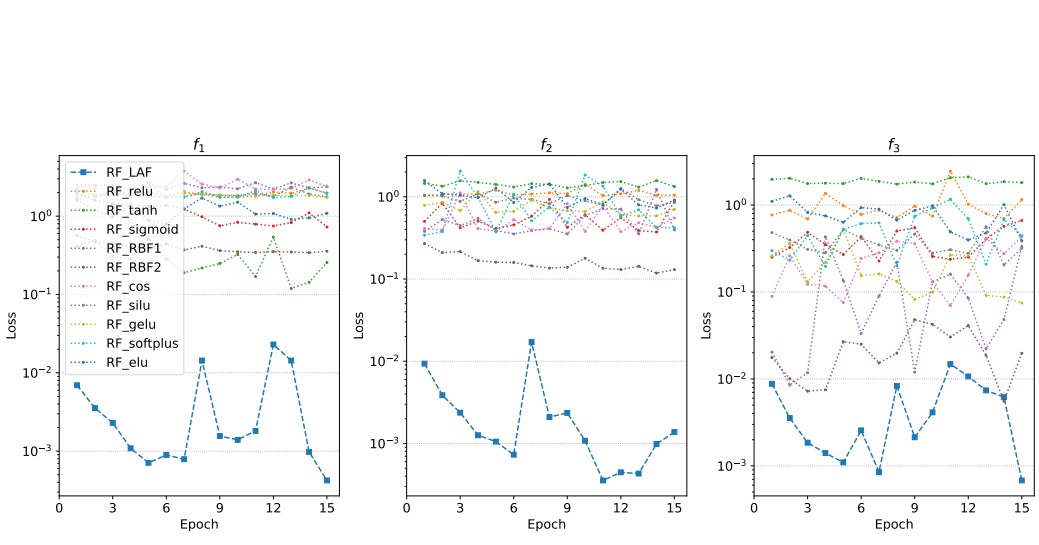

Figure 3: Test losses

