# OpenReview forum: "Random Feature Models with Learnable Activation Functions"
_ICLR.cc/2025/Conference — Submitted to ICLR 2025_

### Official Review · Reviewer_mu4X · 2024-10-28

**Soundness:** 2
**Presentation:** 3
**Contribution:** 2
**Rating:** 3
**Confidence:** 4

**Summary:**

This paper proposes a class of random feature models with activation functions formed by trainable superpositions of radial basis functions.

**Strengths:**

The paper is clearly written, and the mathematics appears correct (though mostly consisting of slight extensions of standard results). However, as I detail below under *Weaknesses*, I'm struggling to muster much enthusiasm for this paper because I think it fails to answer the central question of why one would use these models rather than a standard MLP.

**Weaknesses:**

As mentioned above, I have one major concern with this paper, which overshadows everything else. I will thus be brief. In its current form, the manuscript does not provide any compelling reason why one would use the proposed class of RF models with adaptive activation functions rather than just training the weights (i.e., using an MLP). The experiments provide no comparison against an MLP baseline, and I'm not convinced by the authors' argument that their models will be more interpretable. To understand the nature of signal processing by these networks, one must understand the filtering properties of the random weights as well as the activation function. The theoretical results do not outweigh these concerns, as they are mostly minor modifications of standard RF model analyses. Without any clear comparison to an ordinary neural network, this is for me a clear reject, without further critique required.

**Questions:**

- The discussion after Theorem 5.1 mentions the question of tightness of the bounds on the number of features required in Rudi & Rosasco; the authors might find the corresponding commentary in https://arxiv.org/abs/2405.15699 to be of interest.

---

> ### Author Response · Authors · 2024-11-19
>
> We thank the reviewer for the constructive feedbacks. We also sincerely appreciate the reviewer's recognition of the contribution made by our work. Below we address the central question of the reviewer and would respectfully ask the reviewer to consider increasing the score if our response satisfactorily address the concern.
>
> The central question of the reviewer is why one would use the proposed random feature models rather than a standard MLP. We do have experimental results on the comparison between RFLAF and the counterpart of MLP (RF_relu in Figure 1). RF_relu is also referred to as RFMLP in some paper[1]. And we believe that **comparison between RFLAF and RFMLP instead of a standard MLP is sufficient**. Here are a few reasons.
>
> In this paper, we try to reveal the model properties induced by the component of parametrized activation function through the lens of random feature models. **We do not propose a fundamental neural network structure that challenges MLP.** The model RFLAF is the case of a two-layer Learnable Activation Network (LAN[2], i.e. MLP with learnable activations) where the first-layer parameters are frozen with Gaussian initializations. To understand a component of a neural network better, researchers have attempted to study the random feature version of the corresponding model for more insightful results[1]. Hence we propose to study RFLAF and it is only fair to compare learnable activation networks with MLP both in the random feature settings. The theoretical and experimental results show that parametrized activation component does result in stronger expressivity and interpretability. On the contrary, if the paper is about proposing LAN as a fundamental neural network structure (like KAN[2]), it is necessary to compare LAN (rather than RFLAF) with MLP. However, this is not the intention of our paper.
>
> **Moreover, the properties of RFLAF(randomized LAN) contribute to reflecting the properties of LAN when trained in the lazy regime**. In the training process of a neural network, the model parameters sometimes evolve around the initial parameters, and the model degenerates to a linearized or tangent model (also known for **NTK**, Neural Tangent Kernel). The literature names it the lazy regime of training a neural network. **In the lazy regime, the trained model generalizes approximately the same way as the random feature version of the model (e.g. Proposition A.1 in [3])**. As a result, if one wants to compare the lazy LAN with the lazy MLP, they will also consider comparing RFLAF with RFMLP instead of a standard MLP.
>
> Overall, the theoretical study of the random feature version of a neural network model is important. Not only does it provide insights into the generalization mechanism of a neural network (as illustrated above), but also theoretical results for random features can be leveraged to explain the intriguing phenomena such as *double descent* and the ability to fit random labels in deep neural networks[4-7]. The representation and generalization properties of RFLAF stated and proved in our work are fundamental in both the aforementioned situations.
>
> We believe that the absence of the presentations of the relations between RF models and the theoretical understandings on neural network in the introduction part might have led to such confusions. And we would soon supplement the contents in a revised version. We appreciate the reviewer’s feedback that makes the paper more self-contained.
>
> [1] What can a Single Attention Layer Learn? A Study Through the Random Features Lens
>
> [2] KAN: Kolmogorov–Arnold Networks
>
> [3] On Lazy Training in Differentiable Programming
>
> [4] On exact computation with an infinitely wide neural net
>
> [5] Generalization bounds of stochastic gradient descent for wide and deep neural networks
>
> [6] Fine-grained analysis of optimization and generalization for overparameterized two-layer neural networks
>
> [7] Polylogarithmic width suffices for gradient descent to achieve arbitrarily small test error with shallow ReLU networks

---

> > ### Comment · Reviewer_mu4X · 2024-11-20
> >
> > Thank you for your reply. I'm quite familiar with the literature on random feature models, and do not need to be convinced that studying them is fruitful. However, in those cases there's a clear link between the RF model being studied and the deep network or kernel method one is fundamentally interested in understanding. I don't see such a clear motivation here (for instance, the authors lack a theorem that rigorously establishes a lazy limit for KANs), nor do I find the theoretical contributions significant enough in their own right to justify acceptance at ICLR. I also remain unconvinced by your claims of interpretability. For instance, you've not looked systematically at how the eigenvalues and eigenvectors of the RF kernel change over training of the activations, which once the activations are fixed will determine the model's inductive bias.
> >
> > I also would like to make a small comment regarding your dismissive response to Reviewer Mr13's question about trainable RF approximations to the softmax kernel. I don't think this line of work is as irrelevant as you claim, as it's naturally related to the view of RF models as an approximation to some infinite-rank kernel.

---

> > > ### Author Response · Authors · 2024-11-21
> > >
> > > The main contributions of our work are the representation and generalization theorems. The results are not directly obvious if solely based on current literature, as the model incorporates components of learnable activation functions that introduce intrinsically new properties. The proof of which are also technical (e.g, the solution to the recurrence relation in the proof of theorem 3.2, and the derivation of the concentration property of theorem 3.4). Hence, the mathematics and theorems are not merely slight extensions of standard results. In addition, with the motivation of providing theoretical analyses for the RF model, the contents are complete and adequate for a single paper (as is determined in the review [1]).
> > >
> > > Therefore, we sincerely and humbly ask the reviewer for more detailed reasons on why the theoretical contributions are not significant enough.
> > >
> > > For other questions:
> > > 1. Lacking a clear link between the RF model and the neural network: The literature has already provided a basic relation that the tangent kernel of a network can be decomposed as $K(x,x’) = K^a(x,x’) + K^b(x,x’)$ where the latter term $K^b(x,x’)$ corresponds to the random feature kernel [2]. The link is adequate enough for us to dive deeper into the research of the random feature kernel. And in fact we are considering adding some statements regarding the relations between the two notions in our paper. However, for the case of LAN, a finer analysis between $K$ and $K^b$ remains to be a big open problem. One might need to write another paper to address the problem in the future.
> > > 2. Not considering the dynamics of eigenvalues and eigenvectors: This is another big problem regarding the RFLAF model and should be addressed in a new paper. To study the representation and generalization properties of RFLAF, or to study the eigenvalue dynamics, they are two perspectives of studying the model. We believe that both aspects are insightful and meaningful, and the latter is left for future research. In this paper, we mainly consider the first aspect.
> > >
> > > Finally, we would like to humbly ask the reviewer what exactly is the most significant reason for the clear reject? Does our first response address the reviewer’s initial question of why we did not compare RFLAF with the standard MLP? Does our second response address the reviewer’s subsequent questions? Your reply would be quite precious and meaningful for us.
> > >
> > > We appreciate the reviewer for the inspiring comments. We appreciate any further comments from the reviewer that are related to the main contributions and contents of our paper.
> > >
> > > [1] Random Features for Kernel Approximation: A Survey on Algorithms, Theory, and Beyond
> > >
> > > [2] On Lazy Training in Differentiable Programming

---

> > > > ### Comment · Reviewer_mu4X · 2024-11-22
> > > >
> > > > Thank you for your reply.
> > > >
> > > > 1. Your argument regarding the relationship of the NNGP kernel and the NTK applies in the standard setting of fixed activation functions, but you must show that this also holds if one considers the NTK for trainable activations. This in turn should depend on how you construct a lazy limit; your model would result from an only semi-lazy limit in which the weight matrices are frozen but the activation parameters remain trainable. If you merely adjusted the output scale to get a lazy network, as in Chizat et al 2019, the activation parameters would be frozen along with the weights, and would result in an additional additive component in the NTK formed by the derivatives of the network function with respect to those parameters.
> > > >
> > > > 2. Your response to my question about the eigendecomposition of the trained kernel is very muddled, and I apologize if my question was not clearly motivated. I asked about this analysis because it would give one insight into how the kernel changes with training in a way that directly translates into information about how it will generalize. See, for instance, Canatar, Bordelon, Pehlevan 2021 or Jacot  et al 2020. Seeing how activation functions are reshaped is nice, but measures at the level of the full kernel are necessary.
> > > >
> > > > Also, a point that I failed to mention in my initial review but seems germane to the above: the experiments do not inspire confidence. Why are the epoch-wise learning curves reported in the figures so wildly oscillatory? Moreover, how did you choose the particular RBF kernels used for comparisons?

---

### Official Review · Reviewer_Mr13 · 2024-10-31

**Soundness:** 3
**Presentation:** 2
**Contribution:** 2
**Rating:** 5
**Confidence:** 4

**Summary:**

The paper presents an algorithm for learning activation functions in random feature map representations for neural network models. The authors demonstrate that their learnable variants outperform "static" counterparts, applying fixed nonlinear maps. They also provide theoretical analysis of some of the new random feature map mechanisms they introduce in the paper (Sec. 3: RANDOM FEATURE MODELS WITH A SINGLE RADIAL BASIS FUNCTION). Furthermore, the Authors show that the mechanisms introduced by them needs about twice the parameter number compared to a traditional RF model for substantial quality improvements.

**Strengths:**

- detailed theoretical analysis of the model with a single radial basis function
- an elegant extension of the regular non-learnable random feature map mechanism
- generalization bounds and sample complexity of learning make the theoretical section even more complete

**Weaknesses:**

There are several issues I am concerned about:

1. The paper in general is not very well written. The paper talks about learnable random feature map representations, but from reading the experimental section it is not clear whether the conducted experiments are for MLP layers or activation functions in the linear low-rank attention mechanisms for Transformers (that are mentioned in the related-work part). If the latter is true, I am confused why the comparison does not include also positive random feature map mechanisms that are applied to unbiasedly approximate softmax kernel.

2. The idea of the learnable random feature map representation is not new. In fact, there is a vast literature on positive random feature map mechanisms for the unbiased estimation of the softmax kernel and the most general of those mechanisms do indeed have learnable parameters.

3. The experimental section is very compact. It is really hard to draw any far-reaching conclusions regarding the performance of the method, based on the presented results. Besides, as mentioned above, they are poorly reported.

4. The Authors claim that the mechanisms presented by them introduces about twice the parameter number compared to a traditional RF model for substantial quality gains. This is actually a lot. In the paper it is not reported how this affects speed of training and inference. It is also not clear whether the comparison in the experimental section is with the models of approximately the same of two times smaller number of parameters.

**Questions:**

1. Is the main experimental testbed set up in the attention or MLP setting ?
2. What is the impact on speed of inference and training of the newly introduced parameters ?
3. Can the Authors conduct comprehesive experiments with their models in the attention setting, including also various positive random feature map mechanisms for the unbiased softmax kernel estimation (also the variations with learnable parameters) ?

---

> ### Author Response · Authors · 2024-11-19
>
> We thank the reviewer for the constructive feedbacks and the recognition of the contribution made by our work. Below we address the questions of the reviewer and would respectfully ask the reviewer to consider increasing the score if our response satisfactorily address the concerns.
>
> **Response to the weaknesses:**
>
> 1. This paper (including the related-work part) does not mention anything about attention mechanism or softmax kernel, so I am guessing there are probably some misunderstandings. I am kind of confused about all the arguments regarding linear low-rank attention mechanisms and softmax kernel mentioned hereafter, as they are almost irrelevant to our work. I would appreciate it if the reviewer could provide a further comment or references for better understanding.
> 2. It is true that the idea of the learnable random feature map representation is not new. However, the use of RBF functions as basis for approximation of activation functions is new in the literature. We demonstrate its potential to be applied in more general neural network structure: (1) The convergence of training is possible and fast in the random feature model setting, and has theoretical guarantees[1].  (2) The model shows stronger expressivity and interpretability compared to the standard random feature models. Since activation function is an important component in many modern neural network structure, our work shed light on the possibility of a wider use of parametrized activation functions.
> 3. For more comprehensive validation, we will also test our models on real-world data, and supplement the experimental results in a revised versin. **One of the main motivation of this work is to show the model properties induced by the component of parametrized activation functions.** The tremendously improved representational ability and interpretability are two important points that we illustrate, as already shown in Figure 1 and 2 respectively. And we believe that in the context of random feature model analysis, these illustrations are sufficient. For further exploration of the superiority of the expressivity and interpretability induced by the learnable activation component, one may have to compare LAN (or KAN as is already explored), the non-randomized version of RFLAF,  with MLP. But this is far from our intention.
> 4. **For theoretical analysis**, we consider RFLAF and standard RF models of **the same width**, and the former model shows significantly better expressivity with only about twice the parameter number of the latter one. In the **experimental** setting, however, we constrain all models **to have the same parameter number**. The width of the RFMLP models are approximately doubled, so they should become stronger. Even in this case, RFLAF shows stronger expressivity and interpretability compared to the standard RF models. To respond to the last point of your argument, we do report the information regarding the parameter number in the third paragraph of section 6. About the inference and training speed, we will discuss them in the next part.
>
> **Response to the questions:**
>
> 1. The experimental testbed set up is in the MLP setting. The research topic of this paper is almost irrelevant of the attention mechanism or softmax kernel.
> 2. The inference time complexity of RFLAF is O(NM) compared to O(M) of RFMLP (the standard RF models), where N is the grid number. With the price of increased time complexity, however, the model also gains significant improvement in both representational capacity and interpretability. In fact, quadratic inference time is common in many modern neural network structures such as Transformers and KANs(also contains learnable activations), and is hence proved to be available in practice. After all, it is natural that stronger models require higher time complexity. We believe the analyses of our model are of high theoretical value, and supports for the potential of parametrized activations for wider applications.
> 3. The structure and inductive bias of the multi-layer perceptron and the attention block are way too different, and considering the attention setting would be far from the motivation of our paper, so we would probably not consider adding the attention settings in our paper. However, we do think that the idea of studying the unbiased softmax kernel estimation in the attention mechanism intriguing, and will leave it for future work. We highly appreciate the reviewer for the inspiration.
>
> [1] Nonconvex Optimization Meets Low-Rank Matrix Factorization: An Overview

---

> > ### Comment · Reviewer_Mr13 · 2024-11-25
> > **response**
> >
> > I'd like to thank the Authors for their comments.
> >
> > 1. My concern is that there is some misunderstanding here. In the related work section you **explicitly** mention at least two papers on low-rank linear attention and softmax kernel (l.82). I am not sure how this refers to your statement: "This paper (including the related-work part) does not mention anything about attention mechanism or softmax kernel" from the rebuttal. The general point I want to make here is that MLPs and attention mechanisms are some of the most important computational modules that would benefit from your proposed algorithms and therefore tests involving both are important.
> >
> > 2. I sincerely appreciate Authors' clarification here. It resolves this concern.
> > 3. I do think that the current empirical evaluation is insufficient. The field of random features gives **wide** range of opportunities of large-scale evaluations on tasks that matter a lot for the entire Community. The lack of large scale neural network experiments (in particular Transformers that would benefit a lot from new RF mechanisms) is a red flag. The rebuttal period provides an opportunity to conduct additional experiments and add them to the manuscript. To the best of my knowledge, those experiments were not conducted.
> > 4. Thank you for your comment on the theoretical analysis.
> >
> > The lack of the solid empirical evaluation (including additional speed and inference experiments) makes the exact evaluation of this paper very hard. Since the Authors did not provide additional empirical evidence in the rebuttal period, I would not feel comfortable increasing my score.

---

### Official Review · Reviewer_fA2a · 2024-11-05

**Soundness:** 3
**Presentation:** 3
**Contribution:** 2
**Rating:** 5
**Confidence:** 3

**Summary:**

The authors propose the Random Feature Model with Learnable Activation Functions (RFLAF), a generalization of traditional random feature models. In RFLAF, the activation function is represented as a linear combination of radial basis functions (RBFs) with learnable weights, allowing the model to approximate arbitrary activation functions. The authors provide a closed-form kernel analysis for the case when the activation set consists of a single RBF, and they extend the theoretical analysis to scenarios with multiple RBFs. Experimental results on synthetic datasets demonstrate that RFLAF outperforms random feature models with fixed activation functions.

**Strengths:**

1. Generalization of Random Feature Models: The authors extend traditional random feature models by introducing learnable activation functions, increasing the model’s expressivity and flexibility.
2. Rigorous Mathematical Foundations: The paper provides a thorough theoretical analysis, including derivations of new kernels and bounds on approximation and generalization, which strengthens the model’s credibility.
3. Clear and Accessible Writing: The paper is well-written, with a logical structure and clear explanations that make complex concepts accessible to readers.

**Weaknesses:**

1. Lack of General Closed-Form Solution: While the introduction critiques spline-based models for lacking closed-form analytical kernels (see line 041-046 ), this paper also lacks a closed-form kernel for cases with multiple basis functions. Additionally, despite emphasizing theoretical rigor, the authors rely on the Adam optimizer for learning, suggesting an absence of closed-form solutions in practical application. (Please correct me if I am missing something)

2. Limited Experimental Validation: The model is tested only on synthetic data, with target functions that closely align with the proposed model’s structure. This limited evaluation raises questions about RFLAF’s applicability to real-world scenarios. Validation on more practical datasets, such as tabular data, would strengthen the paper’s claims.

3. Insufficient Exploration of Expressivity and Interpretability: The paper briefly mentions enhanced expressivity and interpretability as benefits of the proposed model, but it does not adequately demonstrate these gains. Showcasing realistic examples would better illustrate the practical advantages in these areas.

**Questions:**

See weaknesses.

---

> ### Author Response · Authors · 2024-11-19
>
> We thank the reviewer for the constructive feedbacks and the recognition of the contribution made by our work. Below we address the questions of the reviewer and would respectfully ask the reviewer to consider increasing the score if our response satisfactorily address the concerns.
>
> **1.1 Lack of General Closed-Form Solution**
>
> The reasons that we did not derive a closed-form solution in the general case are twofold.
>
> First, it is unnecessary to analyze the analytic form of the kernel in the multiple-RBF case. Because the multiple RBFs only act as approximators of the ground-truth activation function. What really matter are (1) the approximation error between the combination of RBFs and the ground-truth activation function, and, if necessary, (2) the analytic form of the kernel induced by the ground truth target function. With the two knowledge together, one may directly estimate the error between the finite-width kernel and the ground truth kernel (See e.g., Theorem 4,5,6 in [1]).
>
> Second, it is essentially difficult in mathematics to obtain the general closed-form solution, because RBFs (even with the same widths and evenly distributed centers) do not have orthogonal properties. Hence, it is probably infeasible to derive some intuitive results. For the case of a single RBF, however, mathematics become pretty and feasible. We discover a new kernel from this, and establish the first analytical results for this kernel. The theoretical results are fundamental, and we believe will always be valuable for future research.
>
> **1.2 Adam for learning**
>
> The optimization problem of learning the model is categorized as the matrix sensing problem, which is well defined and well studied in the literature[2]. Theorem 4 in [2] shows that gradient descent method ensures exponentially fast convergence of the model to the minimizers. And Theorem 4.2 of our paper guarantees the existence of the solutions of arbitrarily small error with sufficiently large $M$. Combining the two results, one can approximate the ground truth solution with arbitrary small error. The convergence is not sensitive to optimizers. It is also fine to use SGD or any other common optimizer.
>
> **2 Limited Experimental Validation**
>
> This is a constructive suggestion. We will supplement experiments on real-world data for more comprehensive validation and provide a revised paper soon.
>
> **3 Insufficient Exploration of Expressivity and Interpretability**
>
> One of the intention of this paper is to reveal the model properties induced by the component of parametrized activation function through the lens of random feature models. The model RFLAF is the case of a two-layer Learnable Activation Network (LAN[3], i.e. MLP with learnable activations) where the first-layer parameters are frozen with Gaussian initializations. To understand a component of a neural network better, researchers have attempted to study the random feature version of the corresponding model for more insightful results[4]. The experimental results of the comparisons among RFLAF, RFMLP and other RF models have demonstrated the stronger expressivity and interpretability (Figure 1 and 2 respectively) resulted from the parametrized activation function. And we have had some synthetic examples to illustrate the interpretability. We believe that in the context of random feature model analysis, these illustrations are sufficient. For further exploration of the superiority of the expressivity and interpretability induced by the learnable activation component, one may have to study LAN (or KAN as is already explored), the non-randomized version of RFLAF, which is far from our intention.
>
> [1] Random Features for Kernel Approximation: A Survey on Algorithms, Theory, and Beyond
>
> [2] Nonconvex Optimization Meets Low-Rank Matrix Factorization: An Overview
>
> [3] KAN: Kolmogorov–Arnold Networks
>
> [4] What can a Single Attention Layer Learn? A Study Through the Random Features Lens

---

> > ### Comment · Reviewer_fA2a · 2024-11-26
> >
> > Thank you for the clarification.
> >
> > > Lack of General Closed-Form Solution
> >
> > I understand the reasoning you provided. However, this part of your paper needs to be revised: "However, the piecewise definition of splines precludes the extraction of a closed-form analytic kernel, hindering the derivation of theoretical insights." Based on your explanation, your method also does not have a closed-form analysis, but there are other ways to extract "theoretical insights." I think the distinction between your approach and prior work is not clearly presented. It may exist, but it hasn't been explained in enough detail.
> >
> > > Limited Experimental Validation
> >
> > I still believe that non-synthetic performance is necessary to properly evaluate the proposed method. Without this, it is difficult to make a meaningful assessment.
> >
> > >Insufficient Exploration of Expressivity and Interpretability
> >
> > The interpretability results on synthetic data are somewhat expected, given the dataset setup and how it aligns with your algorithm design. What kind of interpretability on real data does your approach provide that was not possible with previous methods? I think again I was hoping to see a clear distinction/advantage for interpretability  with your method, which at least has not been presented clearly.

---

### Meta-Review · Area_Chair_VCcG · 2024-12-22

**Metareview:**

This paper investigates random feature maps with learnable activation function, claiming significant gain of expressivity and practical performance. Theoretical formulation, generalization bounds and sample complexity analysis are the strengths of this paper. However, weak experimental section, lack of comparisons to baselines like positive random features, and parameter efficiency were brought up as weaknesses.  Overall, the reviews have consensus that the contributions of this paper do not pass the bar for ICLR acceptance.

**Additional Comments On Reviewer Discussion:**

The weaknesses recognized by multiple reviews on empirical evaluation (including additional speed and inference experiments) was not addressed by the authors during the rebuttal period.  Reviewer mu4X also highlighted lack of clear motivation for the paper (why not learn everything as in an MLP), and unconvincing theoretical contributions and interpretability, making it hard to justify acceptance at ICLR.

---

### Decision · Program_Chairs · 2025-01-22

Reject